

# Realizing triality and $p$-ality by lattice twisted gauging in (1+1)d quantum spin systems

**Da-Chuan Lu[1], Zhengdi Sun[2] and Yi-Zhuang You[1]**

**1** Department of Physics, University of California, San Diego, CA 92093, USA
**2** Mani L. Bhaumik Institute for Theoretical Physics, Department of Physics and Astronomy, University of California Los Angeles, CA 90095, USA

## Abstract

In this paper, we study the twisted gauging on the (1+1)d lattice and construct various non-local mappings on the lattice operators. To be specific, we define the twisted Gauss law operator and implement the twisted gauging of the finite group on the lattice motivated by the orbifolding procedure in the conformal field theory, which involves the data of non-trivial element in the second cohomology group of the gauge group. We show the twisted gauging is equivalent to the two-step procedure of first applying the SPT entangler and then untwisted gauging. We use the twisted gauging to construct the triality (order 3) and $p$-ality (order $p$) mapping on the $\mathbb{Z}_p \times \mathbb{Z}_p$ symmetric Hamiltonians, where $p$ is a prime. Such novel non-local mappings generalize Kramers-Wannier duality and they preserve the locality of symmetric operators but map charged operators to non-local ones. We further construct quantum process to realize these non-local mappings and analyze the induced mappings on the phase diagrams. For theories that are invariant under these non-local mappings, they admit the corresponding non-invertible symmetries. The non-invertible symmetry will constrain the theory at the multicritical point between the gapped phases. We further give the condition when the non-invertible symmetry can have symmetric gapped phase with a unique ground state.

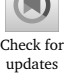

# 1 Introduction

In (1+1)d, Kramers-Wannier duality, Kennedy-Tasaki transformation, and many other duality mappings are powerful tools in solving the quantum spin models [1–7]. These dualities map symmetric operators to other symmetric operators while preserving locality, but they may not preserve locality when mapping charged operators. Consequently, these dualities induce the maps among different gapped and gapless phases [8–14]. They provide a simple and precise way to identify the critical points, understand relatively exotic phases and solve the interacting theories. For the recent discussion of duality mapping and non-invertible symmetry on the lattice, see [15–27].

However, these duality mappings in general lead to the dual theory with a different symmetry. It is difficult to identify them from these non-local mappings. Translating these duality maps to the gauging procedure would make the dual theory and symmetry more explicit [28,29]. Nonetheless, some duality, such as the Kennedy-Tasaki transformation [3–5,30–34], involves "twisted gauging", which will be the main focus of this paper.

In a (1+1)d quantum spin chain, gauging a 0-form global symmetry will lead to a dual 0-form global symmetry [28, 35, 36]. The ordinary Kramers-Wannier duality is obtained by gauging the $\mathbb{Z}_2$ spin-flip symmetry in the transverse field Ising model. In the continuum perspective, for a theory with a global symmetry $G$ together with the 't Hooft anomaly $\omega \in H^3(G, U(1))$, we can gauge its anomaly-free subgroup $H \subseteq G$, twisted by a discrete torsion $\varphi \in H^2(H, U(1))$. The original theory is mapped to a dual theory with the dual (categorical) symmetry $\mathcal{C}(G, \omega, H, \varphi)$, the so-called group-theoretical fusion category, under the twisted

Table 1: Bulk-boundary correspondence between the 2+1d bulk Symmetry TFT and 1+1d boundary theory. The anyon permutation symmetries in the bulk correspond to different transformations on the boundary theory. T corresponds to applying an SPT entangler with non-trivial $\varphi$. S corresponds to gauging with diagonal bicharaters. S followed by R corresponds to gauging with different bicharacters S.

| Bulk anyon permutation symmetry | Transformation on the boundary |
|:---:|:---:|
| T-type | Applying an SPT entangler |
| R-type | Automorphism of global symmetry |
| S-type | Gauging the global symmetry |

gauging. Many physically interesting fusion category symmetries are group-theoretical fusion categories, for example, $\text{Rep}(G) = \mathcal{C}(G, 1, G, 1)$ [20, 25–27] and the Tambara-Yamagami fusion category can be group theoretical with certain conditions [37, 38]. The general gauging of $H$ is specified by:

- a symmetric, non-degenerate bicharacter $\chi : H \times H \to U(1)$, specifying how the dual symmetry should be identified with the original symmetry. For $H = \mathbb{Z}_N$, the identification is unique. But for $H = \mathbb{Z}_N \times \mathbb{Z}_N$, there are diagonal pairing and off-diagonal pairing, which are related by the automorphism of $H$.

- a discrete torsion $\varphi \in H^2(H, U(1))$, corresponding whether applying an SPT entangler before gauging [39].

In the partition function level, the data is presented as[1]

$$\widetilde{Z}[X, \widehat{A}] = \# \sum_{a \in H^1(X, G)} Z[X, a] \exp\left( 2\pi i \int \chi(a, \widehat{A}) + \alpha(a) \right), \tag{1}$$

where $\alpha(g, h) = \varphi(g, h)\varphi(h, g)^{-1}$ and # is the normalization factor. $a$ is the dynamical gauge field being summed over and $\widehat{A}$ is the background gauge field for the dual global symmetry. We incorporate these data in the lattice gauging in this paper, the non-trivial discrete torsion $\varphi(g, h)$ will modify the Gauss law operator (17) and different bicharacters are obtained by certain global symmetries. We denote the untwisted (twisted) gauging as gauging with the trivial (non-trivial) discrete torsion and using the diagonal pairing between the original symmetry and the dual symmetry.[2] We implement the twisted gauging using the modified Gauss law, and show on lattice that

$$\text{Twisted gauging} = \text{Applying SPT entangler then untwisted gauging.} \tag{2}$$

There is a bulk-boundary correspondence between the (un)twisted gauging in (1+1)d theory and anyon permutation symmetry of the bulk (2+1)d topological order [40–54], a lightning review of the bulk-boundary correspondence is given in Appendix D. We will mainly focus on the lattice version of the transformations on the (1+1)d boundary theory. In this language, we can reinterpret the duality as gauging, for instance,

---

[1] We assume $G$ is abelian for simplicity. Gauging the non-abelian symmetry $G$ leads to dual non-invertible $\text{Rep}(G)$ symmetry. The more accurate treatment is to use the topological defect line operators, which will be elaborated in Sec. 2.

[2] More generally, the twisted and untwisted gauging only have a relative difference, we thank Sahand for raising the case about anomaly-free non-on-site symmetry.

- Kramers-Wannier duality = Untwisted gauging $\mathbb{Z}_2$ global symmetry (S).

- Kennedy-Tasaki duality = $(-1)$-Twisted gauging $\mathbb{Z}_2 \times \mathbb{Z}_2$ then applying an SPT entangler $(\mathsf{TS}_1\mathsf{S}_2\mathsf{T}^{-1})$,

where T denotes bulk anyon permutation symmetry corresponding to applying the SPT entangler with a non-trivial element in $H^2(H, U(1))$ [55], and $\mathsf{S}_i$ corresponds to the untwisted gauging of $i$-th symmetry with diagonal pairing. There is an additional elementary transformation R corresponds to the automorphism of the global symmetry, which can be used to change the diagonal pairing to off-diagonal pairing Tab. 1.

Note that the twisted gauging generates quite general non-local mappings, which are beyond order 2 duality. Combining with the global symmetry actions, the non-local mapping is the generator of triality (order 3), $p$-ality (order $p$) and even $K$-ality, where $K$ is a finite group [56]. In particular, this paper studies the lattice version of triality and $p$-ality which is a combination of twisted gauging and global symmetry action. To be specific,

- Triality $\mathsf{Tri}$ = Twisted gauging $\mathbb{Z}_N \times \mathbb{Z}_N$ follow by an automorphism $(\mathsf{R}(U_1)\mathsf{S}_1\mathsf{S}_2\mathsf{T})$,

- $p$-ality $\mathsf{P}$ = $(-2)$-Twisted gauging $\mathbb{Z}_p \times \mathbb{Z}_p$ follow by an automorphism $(\mathsf{R}(U_1)\mathsf{S}_1\mathsf{S}_2\mathsf{T}^{-2})$,

where $U_1 = \begin{pmatrix} 0 & 1 \\ -1 & 0 \end{pmatrix}$ and $\mathsf{R}(U_1)$ changes the diagonal pairing to off-diagonal pairing. To be concrete, for a 1d chain with $\mathbb{Z}_N^{\mathrm{e}}$, $\mathbb{Z}_N^{\mathrm{o}}$ acting on even and odd sites respectively. The automorphism of $\mathbb{Z}_N^{\mathrm{e}} \times \mathbb{Z}_N^{\mathrm{o}}$ specified by $U_1$ is obtained by $TC^{\mathrm{e}}$, where $T$ is the lattice translation symmetry and $C^{\mathrm{e}}$ is the charge conjugation symmetry acting on the even sites. The translation symmetry $T$ effectively swap the two $\mathbb{Z}_N$s. All the non-local mappings are derived in the algebra level and they induce the mapping among the Hamiltonians. For $\mathbb{Z}_N \times \mathbb{Z}_N$ symmetric Hamiltonians that describe gapped phases, triality maps,

$$\mathsf{Tri}: \quad \mathrm{SPT}^0 = \mathrm{SYM} \underset{\overset{\mathrm{SPT}^{N-1}}{\longleftarrow}}{\longrightarrow} \mathrm{SSB} \;, \qquad \mathrm{SPT}^a \to \mathrm{SPT}^{(-a-1)^{-1}} \;, \qquad (3)$$

where $\mathrm{SPT}^a$ denotes the $a$-th $\mathbb{Z}_N \times \mathbb{Z}_N$ SPT, and the disordered phase (SYM) is equivalent to 0-th $\mathbb{Z}_N \times \mathbb{Z}_N$ SPT. Depending on $N$, there could be triality invariant SPTs. For $N$ being prime numbers, the triality invariant $\mathrm{SPT}^a$ is given by the condition $a(a+1) + 1 = 0 \mod N$, which exists for $N = 3$ or $N = 1 \mod 3$. The 3+1d analog of triality is discussed in [57]. We show that the non-invertible triality non-local mapping can be related to invertible $\mathbb{Z}_3$ automorphism of $\mathbb{Z}_N \times \mathbb{Z}_N$ in Sec. 7.1. For instance, let's consider the $\mathbb{Z}_2 \times \mathbb{Z}_2$ symmetric Hamiltonians, the $\mathbb{Z}_2^{\mathrm{e}} \times \mathbb{Z}_2^{\mathrm{o}}$ acts on even and odd sites of the 1d chain respectively,

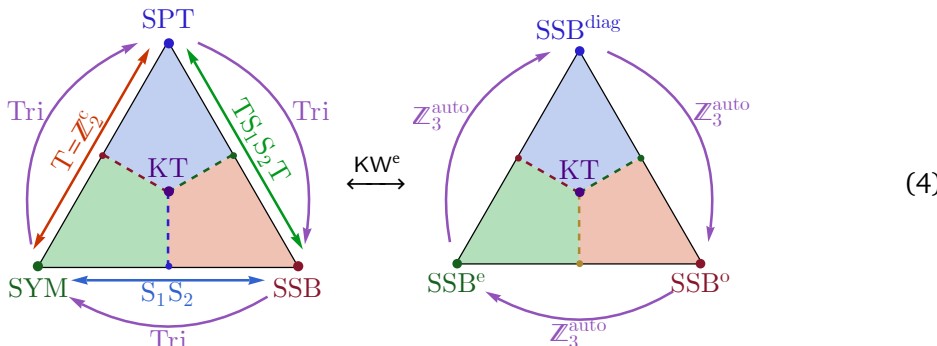

$$(4)$$

where SSB denotes the $\mathbb{Z}_2 \times \mathbb{Z}_2$ spontaneously symmetry breaking phase, and $\mathrm{SSB}^i$ denotes the $i$-th $\mathbb{Z}_2$ partially SSB phase. Under the Kramers-Wannier duality on the even sites, the non-local map Tri is transformed into the $\mathbb{Z}_3^{\mathrm{auto}}$ automorphism of $\mathbb{Z}_2 \times \mathbb{Z}_2$. The center of the

equilateral triangle is the gapless Kosterlitz-Thouless (KT) transition point, which is triality invariant and admits the triality fusion category symmetry.

The $p$-ality transformation acts on the $\mathbb{Z}_p \times \mathbb{Z}_p$ symmetric Hamiltonians with $p$ being a prime number,

$$P: \text{SPT}^1 \circlearrowright \quad \text{SSB} \to \text{SYM} \to \text{SPT}^{2^{-1}} \to \cdots \to \text{SPT}^2 \to \text{SSB}, \qquad \text{SPT}^a \to \text{SPT}^{(-a+2)^{-1}}, \quad (5)$$

where $x^{-1}$ shoud be understood as modular multiplicative inverse of $x$ mod $p$. For $p = 2$ the $p$-ality reduces to the duality with the off-diagonal bicharacter $R(U_1)S_1S_2$ related to the blue arrow in the first diagram of (4), the corresponding non-invertible symmetry can be $\text{Rep}(D_8)$ (one can stack an 2+1d $\mathbb{Z}_2$ SPT to change the Frobenius-Schur indicator of the duality TDL [21] and realize a different fusion category). Similar to the triality, the $p$-ality non-local mapping can be transformed into a $\mathbb{Z}_p^c$ non-on-site symmetry under the $TS_1S_2T^{-1}$ transformation as the green arrow in (4) and in the following for $\mathbb{Z}_3 \times \mathbb{Z}_3$,

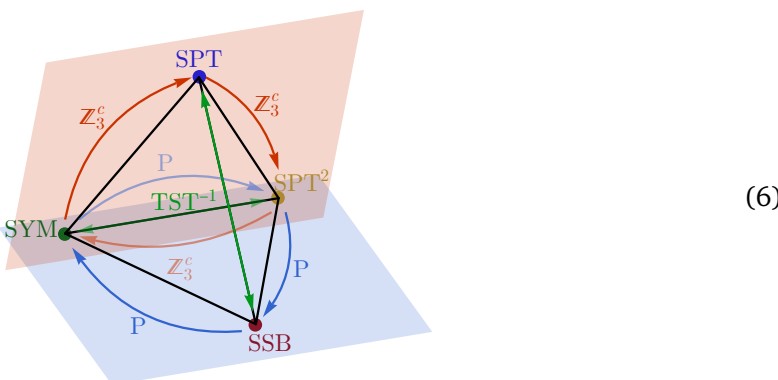

$$(6)$$

The disordered phase (SYM) is equivalent to the $\text{SPT}^0$, the $\mathbb{Z}_3^c$ is generated by the SPT entangler $\prod_j \text{CZ}_{2j-1,2j}^{-1} \text{CZ}_{2j,2j+1}$, where $\text{CZ}_{j,j+1}$ is the controlled-$Z$ gate for qutrits. For $p = 5$,

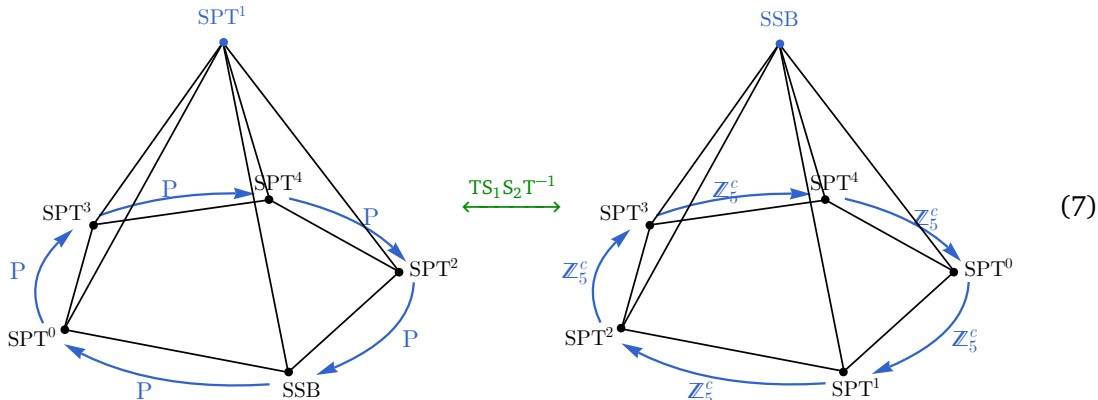

$$(7)$$

The type III mixed anomaly among $\mathbb{Z}_p \times \mathbb{Z}_p \times \mathbb{Z}_p^c$ is used to bootstrap the conformal field theory at the multicritical point between different SPT phases [58], such as the center of the right figure in (7). Under the $TS_1S_2T^{-1}$, the $p$-ality also constrains a multicritical point. For general prime $p$, the $\text{SPT}^1$ is always the $p$-ality invariant phase and admits the $p$-ality fusion category symmetry. It would be interesting to classify the symmetric gapped phases or the fiber functors of the $p$-ality fusion category symmetry [56,59,60]. As we analyzed in Sec. 7.2, the $\text{SPT}^1$ could have several cousins but their domain walls will host zero modes [24,61,62]. More details of these non-local mapping will be elaborated in the main context.

In this paper, we study the non-local mappings on the 1+1d lattice model using the (un)twisted gauging. In particular, we define the twisted Gauss law operator and derive the

dual Hamiltonians under the twisted gauging. We connect the (un)twisted gauging to the quantum process and orbifold in field theory. The locality of symmetric operators is preserved under the (un) twisted gauging, while the charged operators are mapped to non-local ones. The mapping is on the algebra level and does not need to specify the Hamiltonians. To be concrete, we study the triality mapping of $\mathbb{Z}_N \times \mathbb{Z}_N$ symmetric Hamiltonians and $p$-ality mapping of $\mathbb{Z}_p \times \mathbb{Z}_p$ symmetric Hamiltonians in detail, where $p$ is a prime number. Both non-local mappings are generated by the different twisted gauging. We find the condition for triality or $p$-ality invariant gapped phase with a unique ground state. And consequences of the triality or $p$-ality fusion category symmetry. The outline is as follows, we review the gauging in continuum, lattice, and quantum information perspective in Sec. 2 and define the general twisted Gauss law operator on the 1d lattice. In Sec. 3, we review the Kramers-Wannier duality and set up the notations. In Sec. 4, we study the $\mathbb{Z}_2 \times \mathbb{Z}_2$ twisted gauging and construct its corresponding quantum process. By simplifying the steps of minimal coupling and imposing Gauss law, we find that twisted gauging is equivalent to applying the SPT entangler and then gauging. We summarize the non-local mapping among gapped phases in Sec. 4.1. We generalize the twisted gauging of $\mathbb{Z}_N \times \mathbb{Z}_N$ in Sec. 5 and detailed study the triality and $p$-ality non-local mapping in Sec. 6. Specifically, we derive the triality and $p$-ality non-local mappings on symmetric Hamiltonians and find the condition when the symmetric gapped phases with unique ground state are invariant under such mapping. We further study their corresponding fusion category symmetries. In Sec. 7, we give the continuum field theory perspective and connect them to the lattice transformations. We give the group theoretical fusion category construction of the triality and $p$-ality fusion category, which implies that the non-local mappings can be converted into the invertible symmetries.

We will use mathsf font for general (non-)local mapping of the operators in the Hamiltonians and quantum gates. $\mathsf{K}_{\dots}$ represents the corresponding Kraus operator. $\mathsf{T}_{\dots}$ represents the matrix acting on the Pauli polynomials as reviewed in Appendix C. $\mathsf{U}_{\dots}$ represents the corresponding unitaries. mathrm font, like S, T, R, labels the bulk automorphism symmetry in the one higher dimensional symmetry topological field theory as reviewed in Appendix D.

## 1.1 Relation to the previous studies

Another important consequence of the duality is its constraint on the phase diagram. As a textbook application, the Kramers-Wannier duality can pin the Ising critical point given there is only one transition point. Since the duality maps one phase to another, the critical point is invariant under the duality [63]. An interesting scenario is that the second-order transition can be driven to the first-order via a multicritical point by turning on the duality-invariant operators [64, 65]. The self-duality is used to constrain the phase diagram and analyze the critical behavior in higher dimensions and other contexts [66–71]. Note that we will focus on the strong duality (exact duality) that maps between ultraviolet theories, in contrast with the weak duality that relates the infrared phases, see recent review of infrared weak duality web in [72–78]. Given the non-local mapping, it is generally hard to find such duality invariant operators, but the (un)twisted gauging procedure again gives clear transformations on the operators. In particular, the (un)twisted gauging can be mapped to a symplectic transformation acting on the stabilizers [79–81]. The duality invariant operators can be easily found using the algebraic method reviewed in Appendix C. To be specific, all the transformations that map the Pauli operator to the Pauli operator while preserving the commutation relations can be represented as the symplectic transformation. The Pauli operators are represented by vectors in the symplectic vector space Appendix C. By solving for the invariant vectors under the symplectic transformation, one can find the non-local-mapping-invariant operators and construct the Hamiltonians.

If a theory is invariant under the duality, it is said to be self-dual. In the generalized symmetry perspective, the self-dual theory admits the non-invertible symmetry associated with the self-duality [82–85]. For (1+1)d systems, the 0-form symmetry is generated by line operators. In particular, these line operators commute with the energy-momentum tensor and they are topological defect line (TDL) operators. The TDL associated to the self-duality can be understood as the interface between the original theory and the dual theory. Because the theory is self-dual, the duality TDL can be moved freely as a consequence of commuting with the Hamiltonian. However, such duality TDL is not invertible. The intuition is that the TDL maps among gapped phases with different ground state degeneracies (GSDs). The ordinary Kramers-Wannier duality maps the ferromagnetic phase with GSD 2 to the paramagnetic phase with GSD 1 and applies twice to get back to the ferromagnetic phase but only with 1 ground state. Then the fusion of duality TDL gives the projection to the symmetric combination of the SSB groundstates. The Kramers-Wannier duality defect has been extensively studied in [6,7,19,86–91] and can be constructed from the half-gauging procedure [21,57,92,93]. For the Kramers-Wannier-like non-invertible symmetry in higher dimension, see [57,92,94,95].

If the theory is invariant under the twisted gauging, the TDLs that are constructed from half-twisted-gauging generate the corresponding non-invertible symmetry. The fusion category consists of the original invertible line and the new non-invertible TDLs as its simple objects. Note that the data of twisted gauging is not enough to determine the fusion category. Additional data should be provided. For example, the Frobenius-Schur indicator of the duality line in the Tambara-Yamagami fusion category is specified by whether or not stacking a (2+1)d SPT from $H^3(\mathbb{Z}_2, U(1))$ before gauging [38,96]. The $\mathbb{Z}_2$ SPT in (2+1)d is discussed in [97,98]. For triality and $p$-ality fusion category, additional data on symmetry fractionalization is needed [60,99,100]. Nevertheless, sets of the triality and $p$-ality fusion category have group theoretical fusion category construction, which is discussed in detail in Sec. 7. The group theoretical fusion category construction suggests that the non-local mapping Tri or P can be converted into invertible symmetry in the dual theories. This is reminiscent of the construction in [24,101].

Similar to ordinary symmetry, the non-invertible symmetry can be anomalous, but the precise meaning deviates from those of ordinary symmetry [102]. It is possible to gauge the anomalous fusion category symmetry $\mathcal{C}$ by inserting the mesh of the Frobenius algebra object but the anomalous fusion category symmetry $\mathcal{C}$ is not compatible with the symmetric gapped phase with a unique ground state. Gauging the non-invertible symmetry is recently discussed in [100,101,103]. The anomaly of a non-invertible symmetry is the obstruction to a symmetric gapped phase with a unique ground state, and mathematically the fusion category does not admit a fiber functor [52,102,104]. For the anomaly condition of the self-duality TY category and its generalization in higher dimension, see [38,62,96,105,106]. If a non-invertible symmetry is anomaly-free, then it has at least one symmetric gapped phase with a unique ground state. The anomaly free non-invertible symmetry is described by the local fusion category [42]. More interestingly, it could have several non-invertible symmetric gapped phases with a unique ground state, but they cannot be smoothly connected to each other [20,24]. As commented in [24], there is no notion of stacking the non-invertible symmetric gapped phases with a unique ground state and there are no symmetric entangler between the gapped phases.

The (un)twsited gauging relates to the idea of preparing the long-range entangled states from measuring the short-range entangled states [107–109]. Moreover, the Kramers-Wannier duality and other mapping are realized by sequential quantum circuits and quantum process [34,110–115]. In particular, we construct the Kraus operator for the twisted gauging of $\mathbb{Z}_N \times \mathbb{Z}_N$ by viewing it as a quantum process. The Kraus operators of Tri and P can be constructed accordingly. It is interesting to generalize the twisted gauging to higher dimensions and realize the low-depth parity check codes [116]. Related construction of the non-local

mapping using the matrix product operator is discussed in [117–123], bond algebraic method in [22, 124, 125] and bilinear phase map is discussed in [32, 33, 126, 127]. The gauging of generalized symmetry using matrix product operator is formulated in [119].

These non-local mappings in (1+1)d that are generated by (un)twisted gauging corresponds to the anyon permutation symmetries of the (2+1)d symmetry topological field theory (SymTFT) [40–54]. In particular, the (1+1)d theories are the boundary of the (2+1)d SymTFT with different boundary conditions. By examining the condensable algebras, one can classify the (1+1)d gapped or gapless theories [10–12, 14, 46, 128, 129]. We review the corresponding anyon permutation symmetries of triality and $p$-ality in Appendix D.

## 2 Gauging in different perspectives

In this section, we mostly review gauging a global symmetry in field theory, lattice model, and quantum circuit perspectives. The fruitful generalization of symmetry in (1+1)d theory is by analyzing the symmetry operators, which are in general codimension-1 topological defect line (TDL) operators [63, 82], see [130–136] for recent reviews on generalized global symmetry. The TDLs commute with energy-momentum tensor of the conformal field theory and thus commute with the Hamiltonian. The mathematical structure of TDLs is generalized from finite group to fusion category [99], where the TDLs are the simple objects of the fusion category [61, 63, 137].

In the following, we will discuss the TDLs in continuum field theory and lattice theory. We review the gauging in the continuum by inserting algebraic objects. We then follow [138] to discuss the TDLs on the lattice. We define the Gauss law operator for the (un)twisted gauging in (17) and pictorially in (18). The Gauss law operator can be efficiently diagonalized using the algebraic method reviewed in Appendix C. The (un)twisted gauging on the lattice can also be implemented by the quantum process (Kraus operator), which involves adding degrees of freedom, unitary transformation, and measuring out degrees of freedom. Our goal is to explain how to translate the (un)twisted gauging procedure step-by-step to quantum processes and derive the corresponding Kraus operators.

**Gauging in 2d CFT** Gauging the symmetry in 2d CFTs is to insert the mesh of algebraic object in the partition function [28, 139] and resulting in a summation of twisted torus partition function [87, 88, 140]. The algebraic object $\mathcal{A}$ is the direct sum of simple TDLs, together with the fusion junction $\mu \in \text{Hom}_{\mathcal{C}}(\mathcal{A} \otimes \mathcal{A}, \mathcal{A})$ and split junction $\mu^{\vee} \in \text{Hom}_{\mathcal{C}}(\mathcal{A}, \mathcal{A} \otimes \mathcal{A})$. Since the gauged theory should be invariant under different triangulations of the manifold, the fusion and split junctions should be invariant under the various $F$-moves, [141]

$$
\tag{8}
$$

$$\text{(diagram)}$$

and the bubble shrinking,

$$\text{(diagram)} \tag{9}$$

For a generic triangulation of the torus, the gauging is to put the mesh of an algebraic object on the dual graph, the partition function is equal to the minimal mesh after the bubble shrinking and $F$-moves,

$$\text{(diagram)} \tag{10}$$

The algebraic object is used to gauge non-invertible symmetry in 2d CFTs [28,139]. For example, to gauge an anomaly-free finite group $G$, the algebra object $\mathcal{A} = \bigoplus_{g \in G} g$. The fusion and split junctions are given by,

$$\text{(diagram)} \tag{11}$$

According to the consistency condition, $\delta\varphi(g,h,k) = \omega(g,h,k) = 1$. $\varphi(g,h) \in H^2(G, U(1))$ and $\varphi^\vee(g,h) = \varphi(g,h)^{-1}$. Inserting the mesh of the algebra object into the partition function, we obtain,

$$\text{(diagram)} = \frac{1}{|G|} \sum_{\substack{g,h \in G \\ gh=hg}} \frac{\varphi(g,h)}{\varphi(h,g)} \; \text{(diagram)} \;. \tag{12}$$

The gauged partition function depends on the choice of the discrete torsion $\varphi(g,h) \in H^2(G, U(1))$. However, there is no natural choice to favor one discrete torsion over the others, since the notion of SPT order is relative. In the quantum information perspective, the SPT phases are short-range entangled states that can be connected to the trivial product state by finite-depth local unitaries.

For example, $H^2(\mathbb{Z}_2, U(1)) = \mathbb{Z}_1$ is trivial, there is a unique choice of gauging $\mathbb{Z}_2$ global symmetry. But for $\mathbb{Z}_2 \times \mathbb{Z}_2$, $H^2(\mathbb{Z}_2 \times \mathbb{Z}_2, U(1)) = \mathbb{Z}_2$, these result in two different ways of gauging (differed by choice of $\pm$ signs in the following gauged partition function),

$$
\begin{aligned}
Z_{\mathcal{T}/\mathbb{Z}_2 \times \mathbb{Z}_2} = \frac{1}{4}( & Z_{\mathcal{T}}[\mathbb{1}, \mathbb{1}, \mathbb{1}] + Z_{\mathcal{T}}[\mathbb{1}, a, a] + Z_{\mathcal{T}}[\mathbb{1}, b, b] + Z_{\mathcal{T}}[\mathbb{1}, ab, ab] + Z_{\mathcal{T}}[a, \mathbb{1}, a] + Z_{\mathcal{T}}[b, \mathbb{1}, b] \\
& + Z_{\mathcal{T}}[ab, \mathbb{1}, ab] + Z_{\mathcal{T}}[a, a, \mathbb{1}] + Z_{\mathcal{T}}[b, b, \mathbb{1}] + Z_{\mathcal{T}}[ab, ab, \mathbb{1}] \pm Z_{\mathcal{T}}[a, b, ab] \\
& \pm Z_{\mathcal{T}}[a, ab, b] \pm Z_{\mathcal{T}}[b, a, ab] \pm Z_{\mathcal{T}}[b, ab, a] \pm Z_{\mathcal{T}}[ab, a, b] \pm Z_{\mathcal{T}}[ab, b, a]),
\end{aligned}
\tag{13}
$$

where $a, b$ are the symmetry lines for $\mathbb{Z}_2^a \times \mathbb{Z}_2^b$, and the twisted torus partition function is defined as follows,

$$
Z[\mathcal{L}_1, \mathcal{L}_2, \mathcal{L}_3; \mu, \nu](\tau) = \quad \vcenter{\hbox{\includegraphics{fig}}} \quad .
\tag{14}
$$

For TDLs of quantum dimension 1, such as group-like TDLs, their fusion junction is 1 dimensional, such that the labels $\mu, \nu$ will be omitted for this case, as in (13).

**Gauging on the lattice**   The connection between field theory and lattice begins with the identification of topological defect lines in the lattice model. This has been elaborated in [138] as well as the Lieb-Schultz-Mattis anomaly and 't Hooft anomaly of the symmetry on the lattice. We follow [138] to define the topological defect lines (TDLs) on the lattice.

Consider the tensor product Hilbert space $\mathcal{H} = \otimes_j \mathcal{H}_j$. For the on-site symmetry $G$, the symmetry TDL corresponding to $g \in G$ is $U^g = \prod_j U_j^g$, where $U_j^g$ is a unitary operator acting non-trivially on the site $j$. It acts on local operator $\mathcal{O}_j^a$ at site $j$ with representation index $a$ as $(U^g)^{-1} \mathcal{O}_j^a U^g = (U_j^g)^{-1} \mathcal{O}_j^a U_j^g = R(g)_{ab} \mathcal{O}_j^b$. And the Hamiltonian $H = (U^g)^{-1} H U^g$ is invariant under the symmetry action as expected.

In the system with Lorentz invariance, the partition function with TDL $\mathcal{L}$ action $\mathrm{Tr}(\mathcal{L} e^{-\beta H})$ is related to the defect partition function $\mathrm{Tr}_{\mathcal{H}_\mathcal{L}} e^{-\beta H}$ by the modular $S$-transformation, where $\mathcal{H}_\mathcal{L}$ is the defect Hilbert space. For a lattice system, there is no Lorentz invariance, and we define the defect Hamiltonian by acting the TDL on half of the space. To be specific, the defect Hamiltonian with a symmetry defect of $g$ at link $(j, j+1)$ is obtained by,

$$
H_g^{(j-1,j)} \equiv (U_{<j}^g)^{-1} H U_{<j}^g, \qquad \text{where} \qquad U_{<j}^g = \prod_{i=-\infty}^{j-1} U_i^g.
\tag{15}
$$

The symmetry defect is topological, and it can be moved freely without energy cost. The defect moving operator is given by,

$$
\lambda_j^g = (U_{<j}^g)^{-1} U_{<j+1}^g = U_j^g, \qquad (\lambda_j^g)^{-1} H_g^{(j-1,j)} \lambda_j^g = H_g^{(j,j+1)}.
\tag{16}
$$

When moving one defect close to another defect, the fusion junction of TDLs is in general a vector space of dimension $N_{ab}^c$, where $a, b, c$ are the TDLs. There is in general a matrix acting on the fusion junction. For group-like TDLs, the fusion junction is 1-dimensional and there is a

phase ambiguity 2-cocycle $\varphi(g, h)$ associated to the junction. Since the $\varphi(g, h)$ is a 2-cocycle, $\delta\varphi(g, h, k) = 0$, it will not introduce new 't Hooft anomaly. Although such phase commutes with the defect Hamiltonian, we will show the phase $\varphi(g, h) \in H^2(G, U(1))$ could change the Gauss law operator, imposing the corresponding twisted Gauss law corresponds to the twisted gauging.

To gauge the symmetry $G$, we first introduce the degrees of freedom on the links labeled by a group element $g$. $|g\rangle_{j-\frac{1}{2}}$ at the link $(j-1, j)$ denotes the $g$ defect between site $j-1$ and $j$. The gauge transformation operator is defined by,

$$^\varphi G_j^g \equiv \sum_{a,b\in G} \left(\varphi(ag^{-1}, g)^\dagger \left|ag^{-1}\right\rangle \langle a|\right)_{j-\frac{1}{2}} \otimes \lambda_j^g \otimes \left(\varphi(g, b)|gb\rangle \langle b|\right)_{j+\frac{1}{2}}. \tag{17}$$

which is pictorially depicted as,

$$^\varphi G_j^g \equiv \sum_{a,b} \qquad \qquad . \tag{18}$$



Different from the Gauss law operator in the previous literature, this Gauss law operator incorporates the data of $\varphi \in H^2(G, U(1))$. The twisted (untwisted) Gauss law operator refers to non-trivial (trivial) $\varphi(g, h)$. We note that the general Gauss law operators $^\varphi G_j^g$ on different sites commute with each other. The product of all the Gauss law operators act the same as the symmetry operator on the physical Hilbert space on sites $\prod_j {}^\varphi G_j^g |\text{sites}\rangle = \prod_j U_j^g |\text{sites}\rangle$, as it is not necessary to require $\prod_j {}^\varphi G_j^g = \prod_j U_j^g$. When imposing the (un)twisted Gauss law $^\varphi G_j^g = 1$, $\forall j$, we obtain the dual physical space. The (un)twisted gauging refers to the process of introducing the link variable, mimial coupling, and imposing the (un)twisted Gauss law constraint.

Technically, one needs to find a unitary to transform the Gauss law operator to its diagonal form, and other terms in the extended Hamiltonian will be transformed accordingly. This unitary in general is hard to find, so we will use the algebra structure of the Pauli polynomials to find the unitary. This method is used in studying stabilizer code [79–81, 142].

The untwisted gauging corresponds to $\varphi(g, h) = 1$, while the twisted gauging is obtained by taking $\varphi(g, h) \in H^2(G, U(1))$. Note that such modification will not contribute to the 't Hooft anomaly. Although the symmetry operator $U^g$ has a phase ambiguity, such phase ambiguity can not trivialize the phase $\varphi(g, h) \in H^2(G, U(1))$ associated with the fusion vertex of the defect lines. The twisted gauging of $\mathbb{Z}_N \times \mathbb{Z}_N$ is explicitly shown in the following section Sec. 5. We will show that twisted gauging is equivalent to first applying SPT entangler and then untwisted gauging the symmetry.

**Gauging via quantum process** Both the untwisted gauging and twisted gauging involve adding additional link degrees of freedom, extending the Hamiltonian, and imposing the Gauss law constraint. These procedures can be translated to quantum processes, that involve adding ancilla freedoms, performing unitary transformations via quantum circuits, making partial measurements, and then post-selecting measurement outcomes to enforce gauge constraints. Here, the notion of a quantum process refers to a completely positive (CP) map of quantum states that are not necessarily trace-preserving, namely $\rho \to K\rho K^\dagger$ by some Kraus operator $K$. To be precise, given that the post-selection procedure (as a projection) is not trace-preserving,

(non-)invertible symmetries $K$ (as TDLs) are generally implemented as quantum processes, other than quantum channels that are completely positive trace-preserving (CPTP) maps.

Nevertheless, a key ingredient in specifying these quantum processes is the sequential quantum circuit that implements the (majority of) operator mappings. We will explain how to realize the gauging process by quantum gates in the quantum circuit. After compilation and simplification of the circuit structure, the mapping between the original Hamiltonian and the dual Hamiltonian can be achieved efficiently. Once having the quantum circuit, one can find general translation invariant local operators that are invariant under the duality transformations and construct the duality invariant Hamiltonians.

## 3 Warm-up: Kramers-Wannier duality in Ising model as $\mathbb{Z}_2$ gauging

In this section, we review the Kramers-Wannier duality in the Ising model and understand it from field theory, lattice, and quantum process perspectives. This section is meant to set up the notation and review the method, all results in this section are not new. The qubit (spin) operators in the original Hamiltonian are $X_j, Z_j, j \in \mathbb{Z}$, while the dual operators are $\widetilde{X}_{j+\frac{1}{2}}, \widetilde{Z}_{j+\frac{1}{2}}, j \in \mathbb{Z}$.

The Ising model on $1d$ lattice is given by,

$$H_{\text{Ising}} = -\sum_j Z_j Z_{j+1} + g X_j, \tag{19}$$

where $X, Y, Z$ are the Pauli matrices. The $\mathbb{Z}_2$ global symmetry is generated by $U^\eta = \prod_j X_j$, and the local operator $Z_j$ is charged under this $\mathbb{Z}_2$ global symmetry. For simplicity, we will first consider an infinite chain, and we will deal with the boundary conditions later. It is well-known that the Kramers-Wannier duality exchanges the paramagnetic (disordered) and ferromagnetic (ordered) phase, and,

$$\text{KW}: \quad Z_j Z_{j+1} \Rightarrow \widetilde{X}_j, \qquad X_j \Rightarrow \widetilde{Z}_{j-1} \widetilde{Z}_j. \tag{20}$$

Hence, the Ising critical point $g = 1$ is invariant under the Kramers-Wannier duality (20). We will use $\Rightarrow$ for mapping using generic quantum processes and their reserves, and $\rightarrow$ for mapping using unitary operators. Although KW maps the local Hamiltonian to the dual Hamiltonian and preserves locality, it maps the $\mathbb{Z}_2$ charged local operator to the non-local disorder operator,

$$\text{KW}: \quad Z_j \Rightarrow \widetilde{X}_j \widetilde{X}_{j+1} \widetilde{X}_{j+2} \cdots. \tag{21}$$

More drastically, KW maps the symmetry operator $U^\eta = \prod_j X_j \Rightarrow \prod_j \widetilde{Z}_{j-1} \widetilde{Z}_j = 1$. The KW cannot be implemented by a unitary operator. However, the Kramers-Wannier duality is obtained by gauging the global $\mathbb{Z}_2$ symmetry,

$$\text{Kramers-Wannier duality} = \text{untwisted gauging } \mathbb{Z}_2 \text{ global symmetry (S)}. \tag{22}$$

**Lattice perspective** To gauge the $\mathbb{Z}_2$ symmetry on the lattice, we follow the procedure in [138]. We first create many $\mathbb{Z}_2$ defects and then make them dynamic. This enlarges the Hilbert space to include link degrees of freedom. In particular, for every link $(j, j + 1)$, we introduce a local Hilbert space as $j + \frac{1}{2}$ with two states labeling the Ising domain wall degrees of freedom. The extended Hamiltonian becomes,

$$H_{\text{Ising-gauged}} = -\sum_j Z_j \widetilde{Z}_{j+\frac{1}{2}} Z_{j+1} + g X_j. \tag{23}$$

The $\mathbb{Z}_2$ defect moving operator is given by $\lambda_j^g = (X_j)^g$. According to (17), the Gauss law operator is given by,

$$G_j^g = \sum_{a,b} (|a-g\rangle \langle a|)_{j-\frac{1}{2}} \otimes X_j^g \otimes (|g+b\rangle \langle b|)_{j+\frac{1}{2}} = \widetilde{X}_{j-\frac{1}{2}}^g X_j^g \widetilde{X}_{j+\frac{1}{2}}^{-g}. \tag{24}$$

Finally, we impose $G_j = 1$, $\forall j$ to project to the dual Hilbert space. To be specific, we use the unitary,

$$U_{\text{cond}} = \prod_j \mathsf{H}_{j+\frac{1}{2}} \prod_j \mathsf{CZ}_{j-\frac{1}{2},j} \prod_j \mathsf{CZ}_{j,j+\frac{1}{2}}, \tag{25}$$

where $\mathsf{CZ}_{i,j} = \frac{1}{2}(1 + Z_i + Z_j - Z_i Z_j)$ and $\mathsf{H}_j = \frac{1}{\sqrt{2}}(Z_j + X_j)$. $U_{\text{cond}}$ is a finite depth unitary quantum circuit,



$$\tag{26}$$

In our convention, the unitary transformation is taken as $\mathcal{O} \xrightarrow{U} \mathcal{O}' = U^\dagger \mathcal{O} U$. Applying the unitary transformation, the extended Hamiltonian and Gauss law become,

$$H_{\text{Ising-gauged}} = -\sum_j Z_j \widetilde{Z}_{j+\frac{1}{2}} Z_{j+1} + g X_j \xrightarrow{U_{\text{cond}}} \widetilde{H}_{\text{Ising-gauged}} = -\sum_j \widetilde{X}_{j+\frac{1}{2}} + g \widetilde{Z}_{j-\frac{1}{2}} X_j \widetilde{Z}_{j+\frac{1}{2}}, \tag{27}$$

$$G_j = \widetilde{X}_{j-\frac{1}{2}} X_j \widetilde{X}_{j+\frac{1}{2}} \xrightarrow{U_{\text{cond}}} \widetilde{G}_j = X_j. \tag{28}$$

By setting $X_j = 1$, we arrived at the dual Ising model,

$$\widetilde{H}_{\text{Ising}} = -\sum_j \widetilde{X}_{j+\frac{1}{2}} + g \widetilde{Z}_{j-\frac{1}{2}} \widetilde{Z}_{j+\frac{1}{2}}. \tag{29}$$

It can be further mapped back to the original Ising model by shifting the lattice by "$j \to j - \frac{1}{2}$" and sending $g \to 1/g$.

**Quantum process perspective**  In the quantum process perspective, introducing new degrees of freedom on the link and minimal coupling can be formulated as introducing ancilla qubits and entangling with the original qubits, and enforcing the Gauss law constraint can be formulated as making measurement and post-selecting the measurement outcome.

To be concrete, we introduce ancilla qubits on the link with an initial state specified by $\widetilde{Z}_{j+\frac{1}{2}} = 1, \forall j \in \mathbb{Z}$. First, we apply a layer of Hadamard gates,

$$U_{\text{initial}} = \prod_j \mathsf{H}_{j+\frac{1}{2}}, \tag{30}$$

to transform these ancilla qubits to the state of $\widetilde{X}_{j+\frac{1}{2}} = +1, \forall j \in \mathbb{Z}$, which will turn out to be more convenient for performing the gauging procedure via minimal coupling. In the quantum process perspective, gauging means entangling the ancilla qubits to the system as the link (gauge string) degree of freedom with Gauss law constraint.

Next, we aim to construct a sequential quantum circuit that can implement the following maps:

$$X_j \to X_j, \qquad \widetilde{X}_{j+\frac{1}{2}} \to \widetilde{X}_{j-\frac{1}{2}} X_j \widetilde{X}_{j+\frac{1}{2}}, \qquad Z_j \to Z_j \widetilde{Z}_{j+\frac{1}{2}} \widetilde{Z}_{j+\frac{3}{2}} \widetilde{Z}_{j+\frac{5}{2}} \cdots. \tag{31}$$

Physically, the second mapping corresponds to the emergence of a local Gauss law constraint $G_j = +1$ from the ancilla qubit initial state $\widetilde{X}_{j+\frac{1}{2}} = +1$, and the last two mappings correspond to the generation and termination of gauge string at the matter field site, realizing the minimal coupling. Such sequential quantum circuit is given by,

$$U_{\text{gauge}} = \prod_j \text{CX}_{j+\frac{1}{2},j-\frac{1}{2}} \text{CX}_{j+\frac{1}{2},j}, \tag{32}$$

where $\text{CX}_{i,j} = \frac{1}{2}(1 + Z_i + X_j - Z_i X_j)$. Together with the introduction of the ancilla qubit, the quantum circuit $U_{\text{initial}}U_{\text{gauge}}$ in each sequential step can be depicted as,

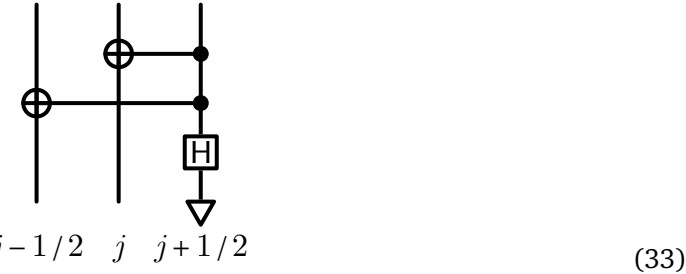

$$\tag{33}$$

Here we stick to the notation that the ancilla qubit (down-triangle) is always introduced in the $Z = +1$ state in the circuit diagram, therefore it requires a Hadamard gate $\text{H}_{j+\frac{1}{2}}$ (as provided by $U_{\text{initial}}$) to transform to the $X = +1$ initial state.

Finally, we need to implement the gauge constraint $G_j = \widetilde{X}_{j-\frac{1}{2}} X_j \widetilde{X}_{j+\frac{1}{2}} = +1, \forall j \in \mathbb{Z}$ by measuring each $G_j$ observable, and post-selecting its measurement outcome to $G_j = +1$. Given that $G_j$ is a non-onsite operator, the first step is to evoke the unitary transformation $U_{\text{cond}}$ in (25) to transform the Gauss law operator $G_j = \widetilde{X}_{j-\frac{1}{2}} X_j \widetilde{X}_{j+\frac{1}{2}}$ to an on-site operator $G'_j = U^\dagger_{\text{cond}} G_j U_{\text{cond}} = X_j$. This amounts to applying the unitary layer $U_{\text{cond}}$ to the system. Then we further apply a final layer of Hadamard gates,

$$U_{\text{final}} = \prod_j \text{H}_j, \tag{34}$$

to transform $X_j \to Z_j$, such that the gauge constraint becomes $Z_j = +1, \forall j \in \mathbb{Z}$ effectively, which can then be implemented by measuring every integer-site qubit in $Z$ basis and post-selecting the $Z = +1$ result.

Note that all these unitary transformations can be combined as a single unitary circuit $U_{\text{KW}}$,

$$\begin{aligned} U_{\text{KW}} &= U_{\text{initial}} U_{\text{gauge}} U_{\text{cond}} U_{\text{final}} \\ &= \prod_j \text{H}_{j+\frac{1}{2}} \prod_j \text{CX}_{j+\frac{1}{2},j-\frac{1}{2}} \text{CX}_{j+\frac{1}{2},j} \prod_j \text{H}_{j+\frac{1}{2}} \prod_j \text{CZ}_{j-\frac{1}{2},j} \prod_j \text{CZ}_{j,j+\frac{1}{2}} \prod_j \text{H}_j \\ &= \prod_j \text{H}_j \text{CX}_{j-\frac{1}{2},j} \text{CX}_{j+\frac{1}{2},j} \text{SWAP}_{j,j+\frac{1}{2}}, \end{aligned} \tag{35}$$

where $\text{SWAP}_{i,j} = \frac{1}{2}(1 + X_i X_j + Y_i Y_j + Z_i Z_j)$. Here we have merged the gates and compiled the quantum circuit into a simpler form.

As a result, the Kramers-Wannier duality KW can be viewed as the composition of introducing of ancilla qubits $\widetilde{Z}_{j+\frac{1}{2}} = 1, \forall j \in \mathbb{Z}$ on the links (half-integer sites), performing the unitary transformation $U_{\text{KW}}$, and post-selecting $Z_j = 1, \forall j \in \mathbb{Z}$ by projective measurement on integer sites. The combined operation goes beyond the scope of unitary transformations. It should be understood as a quantum process implemented by a Kraus operator $K_{\text{KW}}$, such that

any operator mapping $\mathsf{KW} : \mathcal{O} \Rightarrow \mathcal{O}'$ under the Kramers-Wannier duality will be realized as a Kraus map $\mathcal{O}' \propto K_{\mathsf{KW}}^{\dagger} \mathcal{O} K_{\mathsf{KW}}$, or more precisely as $\mathcal{O} K_{\mathsf{KW}} = K_{\mathsf{KW}} \mathcal{O}'$. Thus $K_{\mathsf{KW}}$ should be identified as the duality operator, corresponding to a non-invertible symmetry in the self-dual Ising model.

Inherited from the sequential structure of the unitary quantum circuit $U_{\mathsf{KW}}$, the Kraus operator $K_{\mathsf{KW}}$ also assumes a sequential structure,

$$K_{\mathsf{KW}} = \prod_j K_{j-\frac{1}{2}, j, j+\frac{1}{2}}^{\mathsf{KW}}, \tag{36}$$

where each step of the Kraus operator can be represented by the following quantum circuit diagram following the result of (35),

$$K_{j-\frac{1}{2}, j, j+\frac{1}{2}}^{\mathsf{KW}} = \qquad \qquad \tag{37}$$

$$j - 1/2 \quad j \quad j + 1/2$$

In the above diagram, we assume that the ancilla qubit (down-triangle) is always introduced in the $Z = +1$ state and the single-qubit measurement (square apparatus) is always performed in the $Z$ basis and post-select to the $Z = +1$ outcome. To be more precise, we define the $Z = +1$ projection operator on site-$j$ as

$$\mathsf{P}_j = \frac{1}{2}(1 + Z_j), \tag{38}$$

which enables us to express the Kraus operator in terms of

$$K_{j-\frac{1}{2}, j, j+\frac{1}{2}}^{\mathsf{KW}} = \mathsf{P}_{j+\frac{1}{2}} \mathsf{H}_j \mathsf{CX}_{j-\frac{1}{2}, j} \mathsf{CX}_{j+\frac{1}{2}, j} \mathsf{SWAP}_{j, j+\frac{1}{2}} \mathsf{P}_j, \tag{39}$$

with the understanding that the projection operator $\mathsf{P}_{j+\frac{1}{2}}$ prepares the a new qubit $j + \frac{1}{2}$ to the $\widetilde{Z}_{j+\frac{1}{2}} = +1$ state and the projection operator $\mathsf{P}_j$ post-selects an existing qubit $j$ to the $Z_j = +1$ state. Since the projection $\mathsf{P}_j$ is not invertible, the Kraus operator $K_{\mathsf{KW}}$ as a whole is also not invertible.

From the simplified quantum circuit, we can see that the introduced ancilla qubit in the $Z = +1$ state is never modified by any gate in the circuit until it gets measured to the same $Z = +1$ state with probability one (therefore no selection is actually needed). Effectively, the introduced ancilla will do nothing and then be projected out by measurement, so the combined operations $\mathsf{P}_{j+\frac{1}{2}} \mathsf{CX}_{j+\frac{1}{2}, j} \mathsf{SWAP}_{j, j+\frac{1}{2}} \mathsf{P}_j$ can be dropped together from the circuit, apart from those in the initial and final steps (we will take care of these boundary operations later). The gauging with respect to symmetries encoded into nilpotent fusion categories can be realised in constant depth as discussed in [123]. The sequential unitary circuit within each step is then simplified to:

$$j - 1 \quad j$$

$$\tag{40}$$

It is easy to check that the sequential unitary circuit maps $Z_j Z_{j+1} \rightarrow X_j$ and $X_j \rightarrow Z_{j-1} Z_j$, as expected for the Kramers-Wannier duality.

However, more care should be given to the gate structure near the left and right boundaries, since the first ancilla qubit on the left boundary will be entangled into the system and not be measured until the sequential quantum circuit runs into its right end. To be more specific, we choose $j = 1$ as the starting point of the periodic chain of size $L$, meaning that the site $L + 1$ is equivalent to the site 1. The first ancilla qubit at $\frac{1}{2}$ will be acted by both $K^{\mathsf{KW}}_{\frac{1}{2},1,\frac{3}{2}}$ and $K^{\mathsf{KW}}_{L-\frac{1}{2},L,\frac{1}{2}}$. The complete quantum circuit that implements the Kramers-Wannier duality is

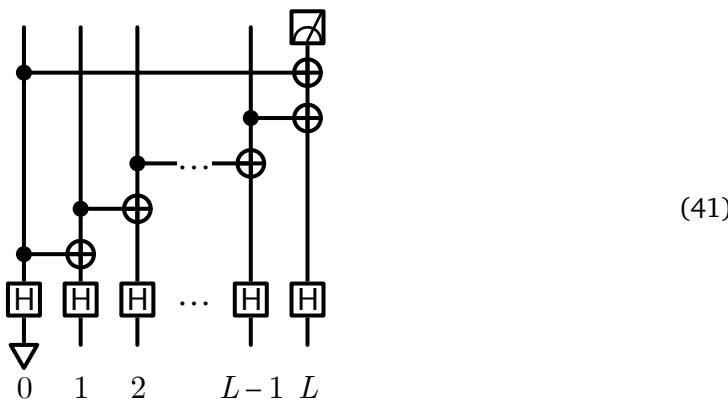

$$(41)$$

where the boundary qubit is relabelled to be $\frac{1}{2} \to 0$. Therefore, we finally arrive at the simplified Kraus operator

$$K_{\mathsf{KW}} = \mathsf{P}_0 \left( \prod_{j=0}^{L} \mathsf{H}_j \prod_{j=1}^{L} \mathsf{CX}_{j-1,j} \right) \mathsf{CX}_{0,L} \mathsf{P}_L \,. \tag{42}$$

In the end, the entire KW quantum process only introduces one ancilla qubit on the left end by $\mathsf{P}_0$, and measures one final qubit on the right end by $\mathsf{P}_L$ [19, 123]. This is reminiscent of orbifolding in the conformal field theory, where the line operators are only inserted once along the non-contractible loops. In particular, the unitary part of $K_{\mathsf{KW}}$ (the quantum circuit part) maps

$$Z_0 Z_L Z_1 \to X_0 \,, \qquad X_L \to Z_L Z_{L-1} Z_0 \,. \tag{43}$$

Therefore, $Z_0$ controls the periodic or anti-periodic boundary condition for the original Ising model, while $Z_L$ controls the boundary condition of the dual model, which are both set to $Z = +1$ by the projection operators $\mathsf{P}_0$ and $\mathsf{P}_L$, realizing periodic boundary conditions in both original and dual Ising models.

Moreover, unitary part of KW maps the symmetry operator as follows,

$$\prod_{k=1}^{L} X_k \to Z_L \,, \qquad Z_0 \to \prod_{k=0}^{L-1} X_k \,. \tag{44}$$

These mappings describe how the original $\mathbb{Z}_2$ symmetry $\prod_{k=1}^{L} X_k$ disappears under gauging and how the dual $\mathbb{Z}_2$ symmetry $\prod_{k=0}^{L-1} X_k$ emerges at the same time. Physically, KW interchanges the symmetry charge and defect charge. For example, let us consider the sector with

$$\prod_{k=1}^{L} X_k = -1 \,, \qquad Z_0 = +1 \,, \tag{45}$$

of the original theory (defined on sites $j = 1, \cdots, L$). Under the KW, it maps to,

$$Z_L = -1 \,, \qquad \prod_{k=0}^{L-1} X_k = +1 \,, \tag{46}$$

which is in the twisted $\mathbb{Z}_2$ even sector of the $\mathbb{Z}_2$ orbifold theory (defined on sites $j = 0, \cdots, L-1$). This is consistent with the $\mathbb{Z}_2$ orbifold of Ising CFT that KW interchanges the sectors,

$$
\begin{array}{c|c|c}
\mathcal{B} & \mathbb{Z}_2\text{-even} & \mathbb{Z}_2\text{-odd} \\
\hline
\text{untwisted} & 1, \epsilon & \sigma \\
\hline
\text{twisted} & \mu & \psi_L, \psi_R
\end{array}
\xleftrightarrow{\ \mathbb{Z}_2\text{-orbifold}\ }
\begin{array}{c|c|c}
\mathcal{B}/\mathbb{Z}_2 & \mathbb{Z}_2\text{-even} & \mathbb{Z}_2\text{-odd} \\
\hline
\text{untwisted} & 1, \epsilon & \mu \\
\hline
\text{twisted} & \sigma & \psi_L, \psi_R
\end{array}
\tag{47}
$$

**Pauli polynomial** In general, it could be hard to keep track of the mapping of all operators under the unitary transformations, as in (25), (32). The algebra method used in quantum stabilizer codes becomes useful for this purpose. As reviewed in Appendix C, the tensor product of Pauli matrices with translation invariance can be represented as a vector, where the coefficients are $\mathbb{Z}_2$-valued. For example, the terms (independent sets of stabilizers) in the Ising model are

$$
Z_j Z_{j+1} \rightsquigarrow \begin{pmatrix} 0 \\ 1+x \end{pmatrix}, \qquad X_j \rightsquigarrow \begin{pmatrix} 1 \\ 0 \end{pmatrix}.
\tag{48}
$$

The unitary transformations will preserve the commutation relation of the stabilizers, and the unitary transformations can be represented as symplectic transformations on the Pauli polynomials. We introduce ancilla qubits with $\widetilde{X}_{j+\frac{1}{2}} = +1, j \in \mathbb{Z}$, which amounts to adding $\widetilde{X}_{j+\frac{1}{2}}$ to each stabilizer set by appending another column of Pauli polynomials in correspondence to $\widetilde{X}_{j+\frac{1}{2}}$,

$$
\begin{pmatrix} 0 \\ 1+x \end{pmatrix} \to \begin{pmatrix} 0 & 0 \\ 0 & 1 \\ 1+x & 0 \\ 0 & 0 \end{pmatrix}, \qquad \begin{pmatrix} 1 \\ 0 \end{pmatrix} \to \begin{pmatrix} 1 & 0 \\ 0 & 1 \\ 0 & 0 \\ 0 & 0 \end{pmatrix},
\tag{49}
$$

where each row now corresponds to $(X_j, \widetilde{X}_{j+1/2}; Z_j, \widetilde{Z}_{j+1/2})$ respectively. The unitary maps $U_{\text{gauge}}$ in (32) and $U_{\text{cond}}$ in (25) are represented as the following symplectic transformations

$$
T_{\text{gauge}} = \begin{pmatrix} 1 & 1 & 0 & 0 \\ 0 & \frac{1}{x}+1 & 0 & 0 \\ 0 & 0 & 1 & 0 \\ 0 & 0 & \frac{1}{x+1} & \frac{1}{x+1} \end{pmatrix}, \qquad T_{\text{cond}} = \begin{pmatrix} 1 & 0 & 0 & 0 \\ 0 & 0 & 0 & 1 \\ 0 & 0 & 1 & 1+x \\ \frac{1}{x}+1 & 1 & 0 & 0 \end{pmatrix}.
\tag{50}
$$

Note that $T_{\text{gauge}}$ is a sequential quantum circuit, while $T_{\text{cond}}$ is a finite depth local unitary circuit, which can be obtained by combining the elementary transformations,

$$
T_{\text{cond}} = T_{\mathsf{CZ}_{j-\frac{1}{2},j}} \cdot T_{\mathsf{CZ}_{j,j+\frac{1}{2}}} \cdot T_{\mathsf{H}_{j+\frac{1}{2}}} = \begin{pmatrix} 1 & 0 & 0 & 0 \\ 0 & 1 & 0 & 0 \\ 0 & x & 1 & 0 \\ \frac{1}{x} & 0 & 0 & 1 \end{pmatrix} \cdot \begin{pmatrix} 1 & 0 & 0 & 0 \\ 0 & 1 & 0 & 0 \\ 0 & 1 & 1 & 0 \\ 1 & 0 & 0 & 1 \end{pmatrix} \cdot \begin{pmatrix} 1 & 0 & 0 & 0 \\ 0 & 0 & 0 & 1 \\ 0 & 0 & 1 & 0 \\ 0 & 1 & 0 & 0 \end{pmatrix}.
\tag{51}
$$

Then the symplectic transformation of KW is

$$
T_{\mathsf{KW}} = T_{\text{cond}} \cdot T_{\text{gauge}} = \begin{pmatrix} 1 & 1 & 0 & 0 \\ 0 & 0 & \frac{1}{x+1} & \frac{1}{x+1} \\ 0 & 0 & 0 & 1 \\ \frac{1}{x}+1 & 0 & 0 & 0 \end{pmatrix}.
\tag{52}
$$

It is straightforward to check the KW transformation on the local operators $X_j, Z_j$ is,

$$
(X_j, Z_j) \rightsquigarrow \begin{pmatrix} 1 & 0 \\ 0 & 0 \\ 0 & 1 \\ 0 & 0 \end{pmatrix} \xrightarrow{T_{\mathsf{KW}}} \begin{pmatrix} 1 & 0 \\ 0 & \frac{1}{1+x} \\ 0 & 0 \\ \frac{1}{x}+1 & 0 \end{pmatrix} \xrightarrow{\text{Gauss law}} \begin{pmatrix} 0 & \frac{1}{1+x} \\ \frac{1}{x}+1 & 0 \end{pmatrix} \leftrightsquigarrow \left( \widetilde{Z}_{j-\frac{1}{2}} \widetilde{Z}_{j+\frac{1}{2}}, \widetilde{X}_{j+\frac{1}{2}} \widetilde{X}_{j+\frac{3}{2}} \cdots \right),
\tag{53}
$$

where the Gauss law is imposed in the last step to set $X_j = +1, j \in \mathbb{Z}$, effectively removing the integer sites from the system. After site relabeling of $j + \frac{1}{2} \to j$, the symplectic transformation

$$T_{\mathsf{KW}}^{\mathrm{eff}} = \begin{pmatrix} 0 & \frac{1}{1+x} \\ \frac{1}{x}+1 & 0 \end{pmatrix}, \tag{54}$$

corresponds to the effective KW transformation, under which the operators transform as

$$(X_j, Z_j) \rightsquigarrow \begin{pmatrix} 1 & 0 \\ 0 & 1 \end{pmatrix} \xrightarrow{T_{\mathsf{KW}}^{\mathrm{eff}}} \begin{pmatrix} 0 & \frac{1}{1+x} \\ \frac{1}{x}+1 & 0 \end{pmatrix} \rightleftharpoons \left( Z_{j-1}Z_j, \prod_{j' \geq j} X_{j'} \right). \tag{55}$$

In the Kraus operator representation, the KW transformation can be formulated as a quantum process, implemented by the Kraus operator in an infinite system $K_{\mathsf{KW}} = \cdots \prod_j \mathsf{H}_j \prod_j \mathsf{CX}_{j-1,j} \cdots$, such that $X_j K_{\mathsf{KW}} = K_{\mathsf{KW}} Z_{j-1} Z_j$ and $Z_j K_{\mathsf{KW}} = K_{\mathsf{KW}} \prod_{j' \geq j} X_{j'}$. This matches the quantum circuit description in (40), after adding boundary terms and projection operators for finite-sized systems.

## 4 Warm-up with a twist: $\mathbb{Z}_2 \times \mathbb{Z}_2$ twisted gauging in quantum spin chain

In this section, we will elaborate on twisted gauging and its connection with "applying an SPT entangler then untwisted gauging". This twisted gauging generates a non-local mapping between local Hamiltonians, in particular, the gapped phases. We will postpone the in-depth discussion of various duality, triality between gapped phases, and their combination later in this section.

To illustrate the twisted gauging on the lattice, we start with the concrete Hamiltonians that respect $\mathbb{Z}_2 \times \mathbb{Z}_2$ symmetry and describe gapped phases. To be specific, the $\mathbb{Z}_2 \times \mathbb{Z}_2$ spontaneous symmetry breaking phase is described by,

$$H_{\mathrm{SSB}} = \sum_j Z_{2j-1}Z_{2j+1} + Z_{2j}Z_{2j+2}, \tag{56}$$

where $X_j, Z_j$ are Pauli matrices. The SSB Hamiltonian has $\mathbb{Z}_2^{\mathrm{e}} \times \mathbb{Z}_2^{\mathrm{o}}$ symmetry, which is generated by $\eta^{\mathrm{e}} = \prod_j X_{2j}, \eta^{\mathrm{o}} = \prod_j X_{2j-1}$ for even (e) and odd (o) sublattices respectively. The $\mathbb{Z}_2^{\mathrm{e}} \times \mathbb{Z}_2^{\mathrm{o}}$ symmetric states include the cluster state in the SPT phase

$$H_{\mathrm{SPT}} = \sum_j Z_{2j-1}X_{2j}Z_{2j+1} + Z_{2j}X_{2j+1}Z_{2j+2}, \tag{57}$$

and the symmetric trivial product state in the disordered phase,

$$H_{\mathrm{SYM}} = \sum_j X_{2j} + X_{2j+1}. \tag{58}$$

**Lattice perspective**  The $\mathbb{Z}_2^{\mathrm{e}} \times \mathbb{Z}_2^{\mathrm{o}}$ defect moving operator is given by $\lambda_j^{(g_1,g_2)} = X_{2j}^{g_1} X_{2j+1}^{g_2}$, where $g_1 = 0, 1$ ($g_2 = 0, 1$) labels the group element of $\mathbb{Z}_2^{\mathrm{e}}$ ($\mathbb{Z}_2^{\mathrm{o}}$). Different from the previous $\mathbb{Z}_2$ case, $\varphi \in H^2(\mathbb{Z}_2^{\mathrm{e}} \times \mathbb{Z}_2^{\mathrm{o}}, U(1)) = \mathbb{Z}_2$ has non-trivial element explicitly given by $\varphi(g, h) = (-1)^{g_1 h_2}$. This non-trivial element cannot be trivialized by redefining the symmetry operator $U^g \to e^{i\nu(g)} U^g$ no matter how the phase factor $\nu(g)$ is chosen. Therefore, the $\mathbb{Z}_2^{\mathrm{e}} \times \mathbb{Z}_2^{\mathrm{o}}$ Gauss law operator has two choices of $\varphi(g, h)$,

$$\varphi(g, h) = 1, \qquad \varphi(g, h) = (-1)^{g_1 h_2}, \tag{59}$$

for $g = (g_1, g_2), h = (h_1, h_2) \in \mathbb{Z}_2^e \times \mathbb{Z}_2^o$. The corresponding untwisted Gauss law operator with $\varphi(g, h) = 1$ is,

$$G_j^e = \widetilde{X}_{2j-\frac{3}{2}} X_{2j} \widetilde{X}_{2j+\frac{1}{2}}, \qquad G_j^o = \widetilde{X}_{2j-\frac{1}{2}} X_{2j+1} \widetilde{X}_{2j+\frac{3}{2}}, \qquad (60)$$

and the twisted Gauss law operator with $\varphi(g, h) = (-1)^{g_1 h_2}$ is given by,

$$^{\mathrm{tw}}G_j^e = \widetilde{X}_{2j-\frac{3}{2}} \widetilde{Z}_{2j-\frac{1}{2}} X_{2j} \widetilde{X}_{2j+\frac{1}{2}}, \qquad ^{\mathrm{tw}}G_j^o = \widetilde{X}_{2j-\frac{1}{2}} \widetilde{Z}_{2j+\frac{1}{2}} X_{2j+1} \widetilde{X}_{2j+\frac{3}{2}}. \qquad (61)$$

The twisted Gauss law operator is directly computed using (17), and the $\mathbb{Z}_N \times \mathbb{Z}_N$ version is derived in Appendix A. Note that the twisted Gauss law operators are dressed by the operator acting on the other $\mathbb{Z}_2$ gauge field, but the twisted Gauss law operators all commute with each other as they should.

The Hamiltonians are also extended to include link variables. The untwisted gauging is similar to the previous $\mathbb{Z}_2$ example. We are focusing on the twisted gauging in the following discussion. The twisted Gauss law operators can be diagonalized by the unitary transformation, whose explicit form is given by the following finite depth local unitary,

$$^{\mathrm{tw}}U_{\mathrm{cond}} = \prod_j \mathsf{CX}_{2j,2j+\frac{1}{2}} \mathsf{CX}_{2j,2j-\frac{3}{2}} \mathsf{CZ}_{2j,2j-\frac{1}{2}} \prod_j \mathsf{CX}_{2j-1,2j-\frac{1}{2}} \mathsf{CX}_{2j-1,2j-\frac{5}{2}} \mathsf{CZ}_{2j-1,2j-\frac{3}{2}} \prod_j \mathsf{H}_{2j-\frac{1}{2}} \mathsf{H}_{2j+\frac{1}{2}}. \qquad (62)$$

Pictorially, it is given by,

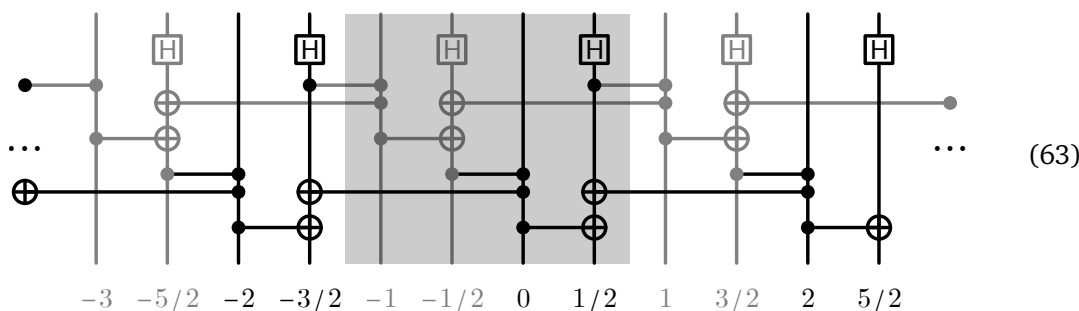

$$(63)$$

where the numbers # correspond to $2j + \#$. The corresponding symplectic transformation is given by,

$$^{\mathrm{tw}}T_{\mathrm{cond}} = \begin{pmatrix} 1 & 0 & 0 & 0 & 0 & 0 & 0 & 0 \\ 0 & 0 & 1 & 0 & 0 & 1 & 0 & 0 \\ 0 & 0 & 1 & 0 & 0 & 0 & 0 & 0 \\ \frac{1}{x} & 0 & 0 & 0 & 0 & 0 & 0 & 1 \\ 0 & 0 & 0 & x & 1 & 1+x & 0 & 0 \\ \frac{1}{x}+1 & 1 & 0 & 0 & 0 & 0 & 0 & 0 \\ 0 & 1 & 0 & 0 & 0 & 0 & 1 & 1+x \\ 0 & 0 & \frac{1}{x}+1 & 1 & 0 & 0 & 0 & 0 \end{pmatrix}, \qquad (64)$$

whose basis is $(X_{2j-1}, \widetilde{X}_{2j-\frac{1}{2}}, X_{2j}, \widetilde{X}_{2j+\frac{1}{2}}, Z_{2j-1}, \widetilde{Z}_{2j-\frac{1}{2}}, Z_{2j}, \widetilde{Z}_{2j+\frac{1}{2}})$. Its action on the gauged stabilizers is,

$$\left( Z_{2j} \widetilde{Z}_{2j+\frac{1}{2}} Z_{2j+2}, Z_{2j-1} \widetilde{Z}_{2j-\frac{1}{2}} Z_{2j+1} \right) \to \left( \widetilde{X}_{2j+\frac{1}{2}}, \widetilde{X}_{2j-\frac{1}{2}} \right), \qquad (65)$$

$$\left( X_{2j}, X_{2j+1} \right) \to \left( \widetilde{Z}_{2j-\frac{3}{2}} \widetilde{X}_{2j-\frac{1}{2}} X_{2j} \widetilde{Z}_{2j+\frac{1}{2}}, \widetilde{Z}_{2j-\frac{1}{2}} \widetilde{X}_{2j+\frac{1}{2}} X_{2j+1} \widetilde{Z}_{2j+\frac{3}{2}} \right), \qquad (66)$$

$$\left( Z_{2j} \widetilde{Z}_{2j+\frac{1}{2}} X_{2j+1} Z_{2j+2}, Z_{2j-1} \widetilde{Z}_{2j-\frac{1}{2}} X_{2j} Z_{2j+1} \right) \to \left( \widetilde{Z}_{2j-\frac{1}{2}} X_{2j+1} \widetilde{Z}_{2j+\frac{3}{2}}, \widetilde{Z}_{2j-\frac{3}{2}} X_{2j} \widetilde{Z}_{2j+\frac{1}{2}} \right). \qquad (67)$$

Its action on the Gauss law operators is as desired,

$$(\widetilde{X}_{2j-\frac{3}{2}} \widetilde{Z}_{2j-\frac{1}{2}} X_{2j} \widetilde{X}_{2j+\frac{1}{2}}, \widetilde{X}_{2j-\frac{1}{2}} \widetilde{Z}_{2j+\frac{1}{2}} X_{2j+1} \widetilde{X}_{2j+\frac{3}{2}}) \to (X_{2j}, X_{2j+1}). \qquad (68)$$

We then impose the Gauss law, $X_j = 1$, and the twisted gauging maps these gapped phases to each other as,

$$\mathsf{TG}: \text{SYM} \xleftrightarrow{\quad\text{SPT}\quad} \text{SSB} \tag{69}$$

For the partial SSB phases,

$$\mathsf{TG}: \text{SSB}^{\text{e}} \leftrightarrow \text{SSB}^{\text{o}}, \qquad \text{SSB}^{\text{diag}} \text{ invariant.} \tag{70}$$

$\mathsf{TG}$ generates an order-6 non-local mapping among gapped phases,

$$(\mathsf{TG})^3 = T, \tag{71}$$

where $T$ is translation by 1 lattice constant (as $j \to j+1$), which will permute the $\mathbb{Z}_2^{\text{e}}$ and $\mathbb{Z}_2^{\text{o}}$ symmetries.

**Quantum process perspective** The minimal coupling between the local fields and the additional ancilla again can be obtained by unitary transformation as $U_{\text{gauge}}$ in the previous $\mathbb{Z}_2$ case. However, the twisted Gauss law operator requires a modified sequential quantum circuit,

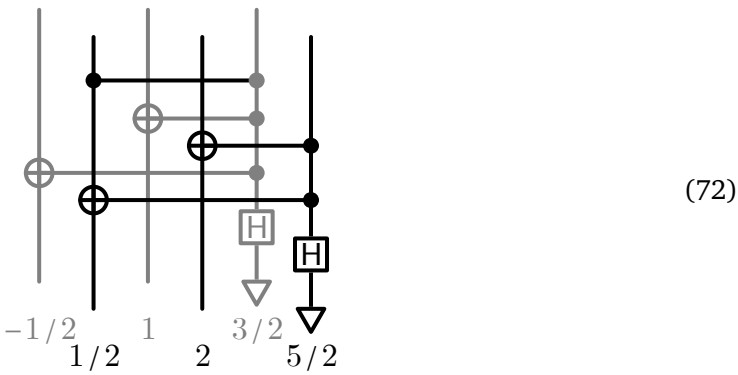

$$\tag{72}$$

where $(-1/2, 1, 3/2) \sim (2j-5/2, 2j-1, 2j-1/2)$ and $(1/2, 2, 5/2) \sim (2j-3/2, 2j, 2j+1/2)$. Its corresponding symplectic transformation is,

$$
^{\text{tw}}T_{\text{gauge}} =
\begin{pmatrix}
1 & 1 & 0 & 0 & 0 & 0 & 0 & 0 \\
0 & \frac{1}{x}+1 & 0 & 0 & 0 & 0 & 0 & 0 \\
0 & 0 & 1 & 1 & 0 & 0 & 0 & 0 \\
0 & 0 & 0 & \frac{1}{x}+1 & 0 & 0 & 0 & 0 \\
0 & 0 & 0 & 0 & 1 & 0 & 0 & 0 \\
0 & 0 & 0 & 1 & \frac{1}{x+1} & \frac{1}{x+1} & 0 & 0 \\
0 & 0 & 0 & 0 & 0 & 0 & 1 & 0 \\
0 & \frac{1}{x} & 0 & 0 & 0 & 0 & \frac{1}{x+1} & \frac{1}{x+1}
\end{pmatrix}. \tag{73}
$$

It is straightforward to check the combined action on local operators, the effective symplectic transformation with Gauss law constraint is,

$$
^{\text{tw}}T_{\text{cond}} \cdot {}^{\text{tw}}T_{\text{gauge}} \xrightarrow{\text{project}}
\begin{pmatrix}
0 & 1 & \frac{1}{1+x} & 0 \\
\frac{1}{x} & 0 & 0 & \frac{1}{1+x} \\
\frac{1}{x}+1 & 0 & 0 & 0 \\
0 & \frac{1}{x}+1 & 0 & 0
\end{pmatrix} \equiv T_{\mathsf{TG}}, \tag{74}
$$

which acts on $(X_{2j-1}, X_{2j}; Z_{2j-1}, Z_{2j})$ basis. It is easy to verify that the symplectic transformation $\mathsf{TG}$ is equivalent to first doing the SPT entangler and then doing Kramers-Wannier duality

on even and odd sites separately,

$$
T_{\text{TG}} = T_{\text{KW}^{\text{e}}} \cdot T_{\text{KW}^{\text{o}}} \cdot T_{\text{SPT}}
$$

$$
= \begin{pmatrix} 0 & 0 & \frac{1}{x+1} & 0 \\ 0 & 1 & 0 & 0 \\ \frac{1}{x}+1 & 0 & 0 & 0 \\ 0 & 0 & 0 & 1 \end{pmatrix} \cdot \begin{pmatrix} 1 & 0 & 0 & 0 \\ 0 & 0 & 0 & \frac{1}{x+1} \\ 0 & 0 & 1 & 0 \\ 0 & \frac{1}{x}+1 & 0 & 0 \end{pmatrix} \cdot \begin{pmatrix} 1 & 0 & 0 & 0 \\ 0 & 1 & 0 & 0 \\ 0 & 1+x & 1 & 0 \\ \frac{1}{x}+1 & 0 & 0 & 1 \end{pmatrix}. \tag{75}
$$

The quantum circuit for the twisted gauging is

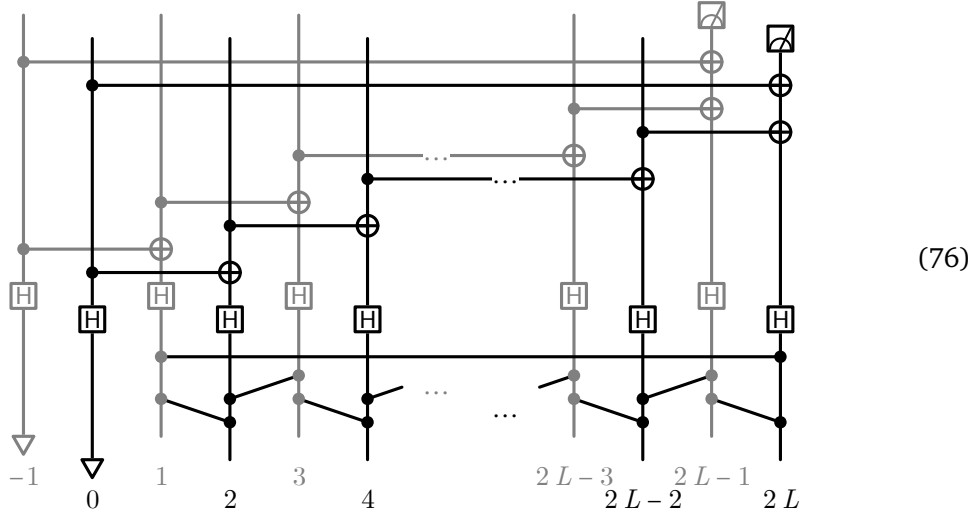

(76)

or expressed as the following Kraus operator

$$
K_{\text{TG}} = U_{\text{SPT}} K_{\text{KW}^{\text{o}}} K_{\text{KW}^{\text{e}}},
$$

$$
U_{\text{SPT}} = \left( \prod_{j=2}^{2L} \text{CZ}_{j-1,j} \right) \text{CZ}_{2L,1},
$$

$$
K_{\text{KW}^{\text{o}}} = \text{P}_{-1} \left( \prod_{j=0}^{L-1} \text{H}_{2j-1} \prod_{j=0}^{L-1} \text{CX}_{2j-1,2j+1} \right) \text{CX}_{-1,2L-1} \text{P}_{2L-1},
$$

$$
K_{\text{KW}^{\text{e}}} = \text{P}_0 \left( \prod_{j=0}^{L-1} \text{H}_{2j} \prod_{j=0}^{L-1} \text{CX}_{2j,2j+2} \right) \text{CX}_{0,2L} \text{P}_{2L}. \tag{77}
$$

The twisted gauging TG already induces a cyclic permutation among the three gap phases: SYM, SPT, and SSB, given that their Hamiltonians are related by the Kraus map:

$$
H_{\text{SYM}} K_{\text{TG}} = K_{\text{TG}} H_{\text{SPT}}, \qquad H_{\text{SPT}} K_{\text{TG}} = K_{\text{TG}} H_{\text{SSB}}, \qquad H_{\text{SSB}} K_{\text{TG}} = K_{\text{TG}} H_{\text{SYM}}. \tag{78}
$$

The slogan for the twisted gauging is,

$$
\text{Twisted gauging} = \text{SPT entangler then gauging } (S_2 S_1 T), \tag{79}
$$

where T corresponds to applying the SPT entangler SPT and $S_1$ ($S_2$) corresponds for the Kramers-Wannier duality $\text{KW}^{\text{e}}$ ($\text{KW}^{\text{o}}$) by gauging each of the two $\mathbb{Z}_2$ symmetries.

We note that the Kennedy-Tasaki (KT) duality is twisted gauging and then applying an SPT entangler. The Kennedy-Tasaki (KT) duality maps $\mathbb{Z}_2 \times \mathbb{Z}_2$ symmetry-protected topological (SPT) phase to $\mathbb{Z}_2 \times \mathbb{Z}_2$ spontaneously symmetry breaking (SSB) phase. The $H_{\text{SYM}}$ is invariant under the KT duality, which is the KT non-invertible symmetry-protected gapped phase.

$$
\text{Kennedy-Tasaki duality} = \text{Twisted gauging } \mathbb{Z}_2 \times \mathbb{Z}_2 \text{ then an SPT entangler } (T S_2 S_1 T). \tag{80}
$$

## 4.1 $(S_3 \times S_3) \rtimes \mathbb{Z}_2$ action on $\mathbb{Z}_2 \times \mathbb{Z}_2$ gapped phases

As discussed in the previous section, the twisted gauging of $\mathbb{Z}_2 \times \mathbb{Z}_2$ is not an order-3 map yet, since $\mathsf{TG}^3 = T$, where $T$ is the lattice translation symmetry, and it will swap the $\mathbb{Z}_2^e$ and $\mathbb{Z}_2^o$. However, under the $T\mathsf{TG}$ map, the SPT phase, symmetric phase and completely SSB phase are permuted, while the partially SSB phases remain invariant,

$$T\mathsf{TG}: \qquad (81)$$

The left and right triangles are related by $\mathsf{KW}^e$.

$$\mathsf{KW}^e: \qquad (82)$$

Moreover, the combination $(\mathsf{KW}^e)^\dagger \circ T\mathsf{TG} \circ \mathsf{KW}^e$ maps,

$$(\mathsf{KW}^e)^\dagger \circ T\mathsf{TG} \circ \mathsf{KW}^e: \qquad (83)$$

In general, the untwisted gauging $\mathsf{KW}^{eo} = \mathsf{KW}^e \circ \mathsf{KW}^o$, together with twisted gauging $\mathsf{TG}$ and $\mathsf{KW}^e$ generate the maps corresponding to the element in $(S_3 \times S_3) \rtimes \mathbb{Z}_2$.[3]

This can be understood from the symmetry topological field theory (SymTFT) perspective. The SymTFT of $\mathbb{Z}_2 \times \mathbb{Z}_2$ symmetric Hamiltonian is $\mathbb{Z}_2 \times \mathbb{Z}_2$ toric code, which contains 16 anyons. The different gapped phases are given by condensing different Lagrangian algebras. The anyons with self-boson statistics can be organized as follows, such that each column and row are mutual bosons,

$$\begin{array}{ccc} e1 & 1e & ee \\ 1m & m1 & mm \\ em & me & ff \end{array} \qquad (84)$$

The table has $(S_3 \times S_3) \rtimes \mathbb{Z}_2$ symmetry, where the $S_3 \times S_3$ permutes the columns and rows, and $\mathbb{Z}_2$ reflects the table along the diagonal (like matrix transpose). This is exactly the anyon permutation symmetry of the $\mathbb{Z}_2 \times \mathbb{Z}_2$ toric code. Depending on condensing which row or column, there are 6 gapped phases,

| Condensible algebra $\mathcal{A}$ | Gapped phase |
|:---:|:---:|
| $11 + e1 + 1e + ee$ | $\mathbb{Z}_2 \times \mathbb{Z}_2$ SSB |
| $11 + 1m + m1 + mm$ | $\mathbb{Z}_2 \times \mathbb{Z}_2$ SYM |
| $11 + em + me + ff$ | $\mathbb{Z}_2 \times \mathbb{Z}_2$ SPT |
| $11 + e1 + 1m + em$ | $\mathbb{Z}_2^e$ SSB |
| $11 + 1e + m1 + me$ | $\mathbb{Z}_2^o$ SSB |
| $11 + ee + mm + ff$ | $\mathbb{Z}_2^{\text{diag}}$ SSB |

$$(85)$$

---

[3]We reserve the term $G$-ality for every elements in $G$ corresponding to (un)twisted gauging [56]. Since in this case the $\mathbb{Z}_2 \times \mathbb{Z}_2$ SPT entangler is a unitary, it differs from other order-2 elements that corresponds to (un)twisted gauging and combining them won't have the proper fusion rule.

The bulk $(S_3 \times S_3) \rtimes \mathbb{Z}_2$ anyon permutation symmetry generates corresponding maps between theories [9],

$$T\mathsf{TG} : \mathsf{SPT} \to \mathsf{SSB} \to \mathsf{SYM} \to \mathsf{SPT}, \tag{86}$$

$$\mathsf{KW}^{\mathrm{eo}} : \mathsf{SYM} \leftrightarrow \mathsf{SSB}, \tag{87}$$

$$(T\mathsf{TG})^\dagger \mathsf{KW}^{\mathrm{eo}} : \mathsf{SPT} \leftrightarrow \mathsf{SYM}, \tag{88}$$

$$(T\mathsf{TG})^\dagger \mathsf{KW}^{\mathrm{eo}}(T\mathsf{TG}) : \mathsf{SPT} \leftrightarrow \mathsf{SSB}, \tag{89}$$

$$\mathsf{KW}^{\mathrm{e}} : \mathsf{SYM} \leftrightarrow \mathbb{Z}_2^{\mathrm{e}} \, \mathsf{SSB}, \qquad \mathsf{SSB} \leftrightarrow \mathbb{Z}_2^{\mathrm{o}} \, \mathsf{SSB}, \qquad \mathsf{SPT} \leftrightarrow \mathbb{Z}_2^{\mathrm{diag}} \, \mathsf{SSB}. \tag{90}$$

If we label the symmetry group as $(S_3^{(1)} \times S_3^{(2)}) \rtimes \mathbb{Z}_2^c$, and the generators satisfy,

$$(a^{(i)})^3 = (b^{(i)})^2 = (b^{(i)}a^{(i)})^2 = 1, \qquad c^2 = 1, \qquad ca^{(i)}c = a^{(\bar{i})}, cb^{(i)}c = b^{(\bar{i})}, \tag{91}$$

where $a^{(i)}, b^{(i)}$ are the generators of $S_3^{(i)}$, and $c$ is the generator for $\mathbb{Z}_2^c$. $\bar{1}, \bar{2} = 2, 1$ respectively. The corresponding maps are related as,

$$T\mathsf{TG} \sim a^{(1)}, \qquad \mathsf{KW}^{\mathrm{eo}} \sim b^{(1)}, \qquad \mathsf{KW}^{\mathrm{e}} \sim c. \tag{92}$$

Then it is clear about the relations of other group elements.

However, the mappings in this $\mathbb{Z}_2 \times \mathbb{Z}_2$ case are not on equal footing. In particular, the element that corresponds to $b^{(1)}a^{(1)}$ is SPT entangler which doesn't involve any gauging, therefore, it corresponds to invertible mapping with quantum dimension 1. In the following, we will specify the $G$-ality as that each element corresponds to (un)twisted gauging, so that it can have a consistent fusion rule. We will discuss triality ($G = \mathbb{Z}_3$) and $p$-ality ($G = \mathbb{Z}_p$) in the subsequent sections. The $S_3$-ality is discussed in [56].

Note that SPT phase is invariant under the Kramers-Wannier duality, while the SYM phase is invariant under the Kennedy-Tasaki transformation. They both admit non-invertible symmetry, and they are corresponding non-invertible symmetric gapped phase with a unique ground state.

## 5 Twisted gauging of $\mathbb{Z}_N \times \mathbb{Z}_N$

It is straightforward to generalize the twisted gauging of $\mathbb{Z}_2 \times \mathbb{Z}_2$ to $\mathbb{Z}_N \times \mathbb{Z}_N$. We first introduce the $\mathbb{Z}_N$ generalized Pauli matrices, also known as shift and clock matrices. Let $X, Z$ be $N \times N$ matrices, acting on the states as $X |n\rangle = |n-1\rangle, Z |n\rangle = \omega^n |n\rangle$, where $\omega = e^{\frac{2\pi i}{N}}$. They satisfy $XZ = \omega ZX$. Using these $\mathbb{Z}_N$ Pauli matrices, we can define controlled gates for the $\mathbb{Z}_N$ case:

$$\mathsf{C}\mathcal{O}_{i,j} = \frac{1}{N} \sum_{a=1}^{N} \sum_{b=1}^{N} \omega^{-ab} Z_i^a \mathcal{O}_j^b, \tag{93}$$

where $i$ is the controlled site and $j$ is the action site, and the operator $\mathcal{O}$ will be replaced by either $X$ or $Z$ to indicate either $\mathsf{CX}$ or $\mathsf{CZ}$ gate. We can also define the projection operator to the $Z = 1$ state (i.e. the state $|n = 0\rangle$) on the site $j$ as

$$\mathsf{P}_j = \frac{1}{N} \sum_{a=1}^{N} Z_j^a. \tag{94}$$

These operators will enable us to construct Kraus operators for $\mathbb{Z}_N \times \mathbb{Z}_N$ twisted gauging.

The $\mathbb{Z}_N \times \mathbb{Z}_N$ SPT is classified by $H^2(\mathbb{Z}_N \times \mathbb{Z}_N, U(1)) = \mathbb{Z}_N$, which is given by,

$$H_{\mathrm{SPT}^a} = \sum_j Z_{2j}^a X_{2j+1} Z_{2j+2}^{-a} + Z_{2j-1}^{-a} X_{2j} Z_{2j+1}^a + h.c. \tag{95}$$

This Hamiltonian is symmetric under the $\mathbb{Z}_N^{\mathrm{e}} \times \mathbb{Z}_N^{\mathrm{o}}$ symmetry which is generated by $\eta^{\mathrm{e}} = \prod_j X_{2j}, \eta^{\mathrm{o}} = \prod_j X_{2j-1}$ on the even (e) and odd (o) sublattices respectively. The SPT Hamiltonian can be obtained from the trivial symmetric phase Hamiltonian $H_{\mathrm{SYM}} \equiv H_{\mathrm{SPT}^0} = \sum_j X_j + h.c.$ by the unitary transformation SPT entangler $U_{\mathrm{SPT}^a} = \prod_j \mathsf{CZ}_{2j-1,2j}^{-a} \mathsf{CZ}_{2j,2j+1}^a$, as $H_{\mathrm{SPT}^a} = U_{\mathrm{SPT}^a}^\dagger H_{\mathrm{SYM}} U_{\mathrm{SPT}^a}$. Here, $\mathrm{SPT}^a$ denotes the $a$th order SPT entangler, which effectively attaches the SPT root state to the system by $a$ times. Another important gapped phase is the spontaneous symmetry breaking phase, which is stabilized by $H_{\mathrm{SSB}} = \sum_j Z_{2j-1} Z_{2j+1}^{-1} + Z_{2j} Z_{2j+2}^{-1} + h.c.$.

The defect moving operator is given by $\lambda_j^{g_1,g_2} = X_{2j}^{g_1} X_{2j+1}^{g_2}$. The phase associated with the defect fusion junction is given by $\varphi(g,h) = \omega^{bg_1h_2} \in H^2(\mathbb{Z}_N \times \mathbb{Z}_N, U(1))$, where every integer $b \in \mathbb{Z}_N$ labels a twisted gauging, called $b$-twisted gauging. Then, following the general formula of (17), the associated $b$-twisted Gauss law operators for $\mathbb{Z}_N^{\mathrm{e}}$ and $\mathbb{Z}_N^{\mathrm{o}}$ are given by

$$^{b}G_j^{\mathrm{e}} = \widetilde{X}_{2j-\frac{3}{2}} \widetilde{Z}_{2j-\frac{1}{2}}^b X_{2j} \widetilde{X}_{2j+\frac{1}{2}}^{-1}, \qquad {}^{b}G_j^{\mathrm{o}} = \widetilde{X}_{2j-\frac{1}{2}} \widetilde{Z}_{2j+\frac{1}{2}}^{-b} X_{2j+1} \widetilde{X}_{2j+\frac{3}{2}}^{-1}. \tag{96}$$

The detailed derivation of the twisted Gauss law operator is in Appendix A. Under the twisted gauging, the stabilizers for the gapped phase become,

$$\left( Z_{2j} Z_{2j+2}^{-1}, Z_{2j-1} Z_{2j+1}^{-1} \right) \rightarrow \left( \widetilde{X}_{2j+\frac{1}{2}}, \widetilde{X}_{2j-\frac{1}{2}} \right), \tag{97}$$

$$\left( X_{2j}, X_{2j+1} \right) \rightarrow \left( \widetilde{Z}_{2j-\frac{3}{2}} \widetilde{X}_{2j-\frac{1}{2}}^{-b} \widetilde{Z}_{2j+\frac{1}{2}}^{-1}, \widetilde{Z}_{2j-\frac{1}{2}} \widetilde{X}_{2j+\frac{1}{2}}^b \widetilde{Z}_{2j+\frac{3}{2}}^{-1} \right), \tag{98}$$

$$\left( Z_{2j}^a X_{2j+1} Z_{2j+2}^{-a}, Z_{2j-1}^{-a} X_{2j} Z_{2j+1}^a \right) \rightarrow \left( \widetilde{Z}_{2j-\frac{1}{2}} \widetilde{X}_{2j+\frac{1}{2}}^{a+b} \widetilde{Z}_{2j+\frac{3}{2}}^{-1}, \widetilde{Z}_{2j-\frac{3}{2}} \widetilde{X}_{2j-\frac{1}{2}}^{-a-b} \widetilde{Z}_{2j+\frac{1}{2}}^{-1} \right). \tag{99}$$

Using the symplectic transformation representation, the $b$-twisted gauging is given by,

$$^{b}T_{\mathrm{gauge}} = \begin{pmatrix} 1 & -1 & 0 & 0 & 0 & 0 & 0 & 0 \\ 0 & 1-\frac{1}{x} & 0 & 0 & 0 & 0 & 0 & 0 \\ 0 & 0 & 1 & -1 & 0 & 0 & 0 & 0 \\ 0 & 0 & 0 & 1-\frac{1}{x} & 0 & 0 & 0 & 0 \\ 0 & 0 & 0 & 0 & 1 & 0 & 0 & 0 \\ 0 & 0 & 0 & -b & \frac{1}{1-x} & \frac{1}{1-x} & 0 & 0 \\ 0 & 0 & 0 & 0 & 0 & 0 & 1 & 0 \\ 0 & \frac{b}{x} & 0 & 0 & 0 & 0 & \frac{1}{1-x} & \frac{1}{1-x} \end{pmatrix}, \tag{100}$$

whose basis is $(X_{2j-1}, \widetilde{X}_{2j-\frac{1}{2}}, X_{2j}, \widetilde{X}_{2j+\frac{1}{2}}, Z_{2j-1}, \widetilde{Z}_{2j-\frac{1}{2}}, Z_{2j}, \widetilde{Z}_{2j+\frac{1}{2}})$. And the unitary transformation that diagonalizes the Gauss law operator is given by,

$$^{b}T_{\mathrm{cond}} = \begin{pmatrix} -1 & 0 & 0 & 0 & 0 & 0 & 0 & 0 \\ 0 & 0 & -b & 0 & 0 & 1 & 0 & 0 \\ 0 & 0 & 1 & 0 & 0 & 0 & 0 & 0 \\ \frac{b}{x} & 0 & 0 & 0 & 0 & 0 & 0 & 1 \\ 0 & 0 & 0 & -bx & -1 & 1-x & 0 & 0 \\ \frac{1}{x}-1 & -1 & 0 & 0 & 0 & 0 & 0 & 0 \\ 0 & -b & 0 & 0 & 0 & 0 & 1 & -1+x \\ 0 & 0 & \frac{1}{x}-1 & -1 & 0 & 0 & 0 & 0 \end{pmatrix}. \tag{101}$$

The twisted gauging is given by,

$$
{}^{b}T_{\text{cond}} \cdot {}^{b}T_{\text{gauge}} \xrightarrow{\text{project}} T_{\text{TG}^b} =
\begin{pmatrix}
0 & -b & \frac{1}{1-x} & 0 \\
\frac{b}{x} & 0 & 0 & \frac{1}{1-x} \\
\frac{1}{x}-1 & 0 & 0 & 0 \\
0 & \frac{1}{x}-1 & 0 & 0
\end{pmatrix},
\tag{102}
$$

which acts on $(X_{2j-1}, X_{2j}, Z_{2j-1}, Z_{2j})$. One can check (97) in the Pauli polynomial representation, the SSB, SYM and SPT stabilizers transform as

$$
\left(
\begin{array}{cccccc}
0 & 0 & 0 & 1 & x & 0 \\
0 & 0 & 1 & 0 & 0 & 1 \\
\hline
0 & 1-x & 0 & 0 & 0 & ax-a \\
1-x & 0 & 0 & 0 & a-ax & 0
\end{array}
\right)
\xrightarrow{T_{\text{TG}^b}}
\left(
\begin{array}{cccccc}
0 & 1 & -b & 0 & 0 & -a-b \\
1 & 0 & 0 & \frac{b}{x} & a+b & 0 \\
\hline
0 & 0 & 0 & \frac{1}{x}-1 & 1-x & 0 \\
0 & 0 & \frac{1}{x}-1 & 0 & 0 & \frac{1}{x}-1
\end{array}
\right).
\tag{103}
$$

As a non-local mapping among $\mathbb{Z}_N \times \mathbb{Z}_N$ symmetric SPT Hamiltonians (and their corresponding phases), $\text{TG}^b$ maps $\text{SPT}^a$ to $\text{SPT}^{(-a-b)^{-1}}$ when $N$ is a prime number, where the inverse should be understood as the modular multiplicative inverse with respect to $N$. One can easily convert the twisted gauging to a quantum process,

$$
\text{TG}^b = \text{KW}^{\text{e}} \circ \text{KW}^{\text{o}} \circ \text{SPT}^b,
\tag{104}
$$

where the $\text{KW}^{\text{e}}/\text{KW}^{\text{o}}$ corresponds to $\mathbb{Z}_N$ Kramers-Wannier duality on even and odd sites, and $\text{SPT}^b$ corresponds to attaching $b$ multiples of the $\mathbb{Z}_N \times \mathbb{Z}_N$ SPT root state. At the level of symplectic transformation of Pauli polynomials, (104) means (multiplication order goes from right to left as the operator mapping reads $v_P \to T \cdot v_P$, where $v_P$ denotes the vector encoding of Pauli operator $P$),

$$
T_{\text{TG}^b} = T_{\text{KW}^{\text{e}}} \cdot T_{\text{KW}^{\text{o}}} \cdot T_{\text{SPT}^b},
$$

$$
T_{\text{KW}^{\text{e}}} =
\begin{pmatrix}
1 & 0 & 0 & 0 \\
0 & 0 & 0 & \frac{1}{1-x} \\
0 & 0 & 1 & 0 \\
0 & \frac{1}{x}-1 & 0 & 0
\end{pmatrix},
$$

$$
T_{\text{KW}^{\text{o}}} =
\begin{pmatrix}
0 & 0 & \frac{1}{1-x} & 0 \\
0 & 1 & 0 & 0 \\
\frac{1}{x}-1 & 0 & 0 & 0 \\
0 & 0 & 0 & 1
\end{pmatrix},
\tag{105}
$$

$$
T_{\text{SPT}^b} =
\begin{pmatrix}
1 & 0 & 0 & 0 \\
0 & 1 & 0 & 0 \\
0 & -b(1-x) & 1 & 0 \\
b(\frac{1}{x}-1) & 0 & 0 & 1
\end{pmatrix}.
$$

At the level of the Kraus operator, (104) is explicitly realized as (multiplication order goes from left to right as the operator mapping is given by $\mathcal{O} \Rightarrow K^{\dagger}\mathcal{O}K$)

$$
K_{\text{TG}^b} = U_{\text{SPT}^b} K_{\text{KW}^{\text{o}}} K_{\text{KW}^{\text{e}}},
$$

$$
U_{\text{SPT}^b} = \left( \prod_{j=1}^{L} \text{CZ}_{2j-1,2j}^{-b} \text{CZ}_{2j,2j+1}^{b} \right) \text{CZ}_{2L,1}^{b},
$$

$$
K_{\text{KW}^{\text{o}}} = \text{P}_{-1} \left( \prod_{j=0}^{L-1} \text{H}_{2j-1} \prod_{j=0}^{L-1} \text{CX}_{2j-1,2j+1} \right) \text{CX}_{-1,2L-1} \text{P}_{2L-1},
\tag{106}
$$

$$
K_{\text{KW}^{\text{e}}} = \text{P}_0 \left( \prod_{j=0}^{L-1} \text{H}_{2j} \prod_{j=0}^{L-1} \text{CX}_{2j,2j+2} \right) \text{CX}_{0,2L} \text{P}_{2L}.
$$

Under the Kraus map of $\mathsf{TG}^b$, the SPT Hamiltonians with prime $N$ (95) are related by

$$H_{\mathrm{SPT}^a} K_{\mathsf{TG}^b} \sim K_{\mathsf{TG}^b} H_{\mathrm{SPT}^{(-a-b)-1}}\,, \tag{107}$$

where $\sim$ denotes that the Hamiltonians on both sides are in the same SPT phase.

# 6 Non-local mapping among gapped phases with $\mathbb{Z}_N \times \mathbb{Z}_N$ symmetry

In this section, we use the (un)twisted gauging to study the nonlocal mapping among different gapped phases. For $\mathbb{Z}_N \times \mathbb{Z}_N$ symmetric Hamiltonians, the possible low energy gapped phases are disorder phase (SYM = SPT$^0$), symmetry-protected topological phases (SPT$^a$) and spontaneously symmetry breaking phases (SSB). We will consider the $b = 1$ and $b = -2$ twisted gauging. The twisted gauging combined with global symmetry transformation generates the triality and $p$-ality ($p$ is a prime number) mapping between gapped phases.

## 6.1 $\mathsf{Tri} = TC^e \mathsf{TG}^1$ as the triality map

For general $N > 2$, the twisted gauging with $b = 1$ maps,

$$\mathrm{SPT}^0 = \mathrm{SYM} \xleftarrow{\hspace{1.5cm}} \overset{\mathrm{SPT}^{N-1}}{\diagup\hspace{0.5cm}\diagdown} \mathrm{SSB}\,, \tag{108}$$

which looks like an order-3 triality map. However, these 3 gapped phases are symmetric under the $TC^e$ symmetry, where $T$ is the translation and $C^e$ is the charge conjugation on the even sites. If considering partially SSB phases, the $\mathsf{TG}^1$ generates an order-12 map. In particular,

$$(\mathsf{TG}^1)^3 = TC^e\,. \tag{109}$$

Note that the translation symmetry effectively swaps the even and odd $\mathbb{Z}_N^e \times \mathbb{Z}_N^o$ symmetry. In particular, under the $(\mathsf{TG}^1)^3$ map, the partially symmetry breaking phase becomes,

$$H_{\mathrm{SSB}^e} = \sum_j Z_{2j}^a Z_{2j+2}^{-a} + X_{2j+1} + h.c. \xrightarrow{(\mathsf{TG}^1)^3} H_{\mathrm{SSB}^o} = \sum_j Z_{2j-1}^{-a} Z_{2j+1}^a + X_{2j} + h.c. \tag{110}$$

It is straightforward to modify the order-12 non-local mapping to an order-3 triality map $\mathsf{Tri}$ by combining the $TC^e$ symmetry action to $\mathsf{TG}^1$. Since the symmetry action $TC^e$ commutes with the twisted gauging $\mathsf{TG}^1$,

$$\mathsf{Tri} \equiv TC^e \mathsf{TG}^1\,, \qquad (\mathsf{Tri})^3 = (TC^e \mathsf{TG}^1)^3 = \mathbb{1}\,. \tag{111}$$

This can also be seen from the Pauli polynomial representation acting on $(X_{2j-1}, X_{2j}, Z_{2j-1}, Z_{2j})$,

$$T_{\mathsf{Tri}} = \begin{pmatrix} -1 & 0 & 0 & \frac{x}{x-1} \\ 0 & -1 & \frac{1}{1-x} & 0 \\ 0 & -1+x & 0 & 0 \\ \frac{1}{x}-1 & 0 & 0 & 0 \end{pmatrix}, \tag{112}$$

and $T_{\mathsf{Tri}}^3 = 1$. According to (97), the twisted gauging $\mathsf{TG}^1$ combining with the $TC^e$ symmetry action maps the stabilizers of SPTs as,

$$\mathsf{Tri} = TC^e \mathsf{TG}^1 : \left( Z_{2j}^a X_{2j+1} Z_{2j+2}^{-a}, Z_{2j-1}^{-a} X_{2j} Z_{2j+1}^a \right) \to \left( \widetilde{Z}_{2j} \widetilde{X}_{2j+1}^{-a-1} \widetilde{Z}_{2j+2}^{-1}, \widetilde{Z}_{2j-1}^{-1} \widetilde{X}_{2j}^{-a-1} \widetilde{Z}_{2j+1} \right). \tag{113}$$

Table 2: Non-local mapping Tri acts on the gapped phases. For $N = 3$ and $N = 1$ mod 3, there are symmetric gapped phases invariant under Tri. The other gapped phases are permuted by Tri in disjoint orbits.

| $N$ | $G$ | Tri $= TC^e TG^1$ mapping among the gapped phases |
|---|---|---|
| 2 | $\mathbb{Z}_2 \times \mathbb{Z}_2$ | SPT over SYM ← SSB |
| 3 | $\mathbb{Z}_3 \times \mathbb{Z}_3$ | SPT$^2$ over SYM ← SSB, SPT$^1$ ↺ |
| 5 | $\mathbb{Z}_5 \times \mathbb{Z}_5$ | SPT$^4$ over SYM ← SSB, SPT$^2$ over SPT$^1$ ← SPT$^3$ |
| 7 | $\mathbb{Z}_7 \times \mathbb{Z}_7$ | SPT$^6$ over SYM ← SSB, SPT$^3$ over SPT$^1$ ← SPT$^5$, SPT$^2$ ↺, SPT$^4$ ↺ |

For $N$ is prime number, any number $x$ with $1 \le x \le N-1$, has the greatest common divisor $(x, N) = 1$, therefore, $x$ has unique inverse $x^{-1}$, such that $x^{-1}x = 1 \mod N$. The stabilizers after mapping are equivalent to,

$$\left( \widetilde{Z}_{2j}^{(-a-1)^{-1}} \widetilde{X}_{2j+1} \widetilde{Z}_{2j+2}^{-(-a-1)^{-1}}, \widetilde{Z}_{2j-1}^{-(-a-1)^{-1}} \widetilde{X}_{2j} \widetilde{Z}_{2j+1}^{(-a-1)^{-1}} \right), \tag{114}$$

where $x^{-1}$ is understood as the modular multiplicative inverse of $x$ modulo $N$. It is obvious that there is an order-3 map among

$$\text{Tri}: \quad \text{SPT}^0 = \text{SYM} \xleftarrow{\quad \text{SPT}^{N-1} \quad} \text{SSB}. \tag{115}$$

The mapping among other SPTs depends on $a$ and $N$,

$$\text{Tri}: \text{SPT}^a \to \text{SPT}^{(-a-1)^{-1}}. \tag{116}$$

It is interesting to notice that, the SPT$^a$ is invariant under the triality if and only if,

$$a(a+1) + 1 = 0 \mod N. \tag{117}$$

We will consider $N$ as a prime number in the following. The equation has solutions if $N = 1$ mod 3 for general prime $N > 3$. Since (117) is a quadratic equation, it has at most two solutions. If the two solutions are $a_1, a_2$, then by Vieta's relations,

$$a_1 a_2 = 1 \mod N, \qquad a_1 + a_2 = -1 \mod N. \tag{118}$$

For example, $N = 3$, the SPT$^1$ is invariant under the triality. For $N = 7$, both SPT$^2$ and SPT$^4$ are invariant under the triality. More examples of the triality invariant SPTs are summarized in Tab. 2. The first few Tri-ality invariant SPTs are given as follows,

$$\{\text{SPT}_3^1\}, \quad \{\text{SPT}_7^2, \text{SPT}_7^4\}, \quad \{\text{SPT}_{13}^3, \text{SPT}_{13}^9\}, \quad \{\text{SPT}_{19}^7, \text{SPT}_{19}^{11}\}, \quad \{\text{SPT}_{31}^5, \text{SPT}_{31}^{25}\}, \tag{119}$$

where the SPT phases are labelled by SPT$_N^a$. Other $\mathbb{Z}_N \times \mathbb{Z}_N$ SPTs are permuted by order 3 cycles under the Tri-ality map,

| $N$ | 5 | 7 | 11 | 13 |
|---|---|---|---|---|
| Cycles | $(1,2,3)$ | $(1,3,5)$ | $(1,5,9)(2,7,4)(3,8,6)$ | $(1,6,11)(2,4,5)(7,8,10)$. |

$$\tag{120}$$

To summarize, the gapped phases of $\mathbb{Z}_N \times \mathbb{Z}_N$ are invariant under Tri if and only if $N = 3$ or $N = 1 \mod 3$, and for the latter case, there are at least 2 gapped SPTs that are Tri invariant. The Tri maps other gapped phases by order 3 permutations as the consequences of Tri being a triality map. The Tri invariant theories admit triality fusion category symmetry. More detailed analysis of triality fusion category symmetry protected topological phases relies on the study of its fiber functors and we leave it for future study.

The ordinary SPTs cannot be smoothly connected without breaking the protecting symmetry. Similarly, the triality fusion category symmetry-protected topological phases will undergo a phase transition between them, the critical theory can also be triality fusion category symmetric. For example, when $N = 7$,

$$H_{\text{tri}}^7 = \sum_j \lambda(Z_{2j}^2 X_{2j+1} Z_{2j+2}^{-2} + Z_{2j-1}^{-2} X_{2j} Z_{2j+1}^2) + (1-\lambda)(Z_{2j}^4 X_{2j+1} Z_{2j+2}^{-4} + Z_{2j-1}^{-4} X_{2j} Z_{2j+1}^4) + h.c. \quad (121)$$

The transition occurs at $\lambda = \frac{1}{2}$, which is pinned by the duality transformation $T\mathsf{KW}^e\mathsf{KW}^o$. The whole phase diagram is invariant under the triality and admits triality fusion category symmetry. In particular, the non-invertible line operators $\mathcal{Q}$ that corresponds to $\mathsf{Tri} = TC^e\mathsf{TG}^1$ with quantum dimension $N$ has the fusion rules with invertible lines in $\mathbb{Z}_N \times \mathbb{Z}_N$,

$$\mathcal{Q} \otimes \overline{\mathcal{Q}} = \bigoplus_{g \in \mathbb{Z}_N \times \mathbb{Z}_N} g, \qquad g \otimes \mathcal{Q} = \mathcal{Q} \otimes g = \mathcal{Q}, \qquad g \otimes h = gh, \quad (122)$$

$$g \otimes \overline{\mathcal{Q}} = \overline{\mathcal{Q}} \otimes g = \overline{\mathcal{Q}}, \qquad \mathcal{Q} \otimes \mathcal{Q} = N\overline{\mathcal{Q}}, \quad (123)$$

where $g, h \in \mathbb{Z}_N \times \mathbb{Z}_N$ and thus $\mathcal{Q} \otimes \mathcal{Q} \otimes \mathcal{Q} = N \bigoplus_{g \in \mathbb{Z}_N \times \mathbb{Z}_N} g$. The details of this triality fusion category will discussed in Sec. 7. The quantum dimension of $\mathcal{Q}$ can also be counted by the Kraus operator that implements such triality transformation. The $\mathsf{KW}^{eo}$ effectively adds 1 lattice site which corresponds to quantum dimension $N$, while translation $T$, charge conjugation $C^e$ and SPT are unitary transformations that will not change the quantum dimension of the defect line.

## 6.2 $\;\mathsf{P} = TC^e\mathsf{TG}^{-2}$ as the $p$-ality map for $\mathbb{Z}_p \times \mathbb{Z}_p$ with prime $p$

Another special non-local map is an $p$-ality map, which is given by,

$$\mathsf{P} = TC^e\mathsf{TG}^{-2} : T_{\mathsf{P}} = \begin{pmatrix} 2 & 0 & 0 & \frac{x}{x-1} \\ 0 & 2 & \frac{1}{1-x} & 0 \\ 0 & x-1 & 0 & 0 \\ \frac{1}{x}-1 & 0 & 0 & 0 \end{pmatrix}. \quad (124)$$

One can find its action on the stabilizers in the Hamiltonian straightforwardly. Interestingly, $n$ times P transformation yields,

$$(T_{\mathsf{P}})^n = \begin{pmatrix} n+1 & 0 & 0 & \frac{nx}{x-1} \\ 0 & n+1 & \frac{n}{1-x} & 0 \\ 0 & n(x-1) & -n+1 & 0 \\ n(\frac{1}{x}-1) & 0 & 0 & -n+1 \end{pmatrix}. \quad (125)$$

Therefore, for $\mathbb{Z}_p \times \mathbb{Z}_p$ system with prime $p$, $\mathsf{P}^p = \mathbb{1} \mod p$ and P generates an order $p$ non-local mapping. The $p$-ality P maps the SPT phases as,

$$\mathsf{P} : \text{SPT}^a \to \text{SPT}^{(-a+2)^{-1}}, \quad (126)$$

Table 3: Non-local mapping P acts on the $\mathbb{Z}_p \times \mathbb{Z}_p$ gapped phases. For all prime numbers $p$, SPT$^1$ is P invariant. The other $p$ gapped phases are permuted under P.

| $p$ | $G$ | $p$-ality P $= TC^e$TG$^{-2}$ mapping among the gapped phases |
|-----|-----|-------------------------------------------------------------|
| 2 | $\mathbb{Z}_2 \times \mathbb{Z}_2$ | SPT $\circlearrowright$    SSB $\leftrightarrow$ SYM |
| 3 | $\mathbb{Z}_3 \times \mathbb{Z}_3$ | SPT$^1$ $\circlearrowright$    SSB $\to$ SYM $\to$ SPT$^2 \to$ SSB |
| 5 | $\mathbb{Z}_5 \times \mathbb{Z}_5$ | SPT$^1$ $\circlearrowright$    SSB $\to$ SYM $\to$ SPT$^3 \to$ SPT$^4 \to$ SPT$^2 \to$ SSB |
| 7 | $\mathbb{Z}_7 \times \mathbb{Z}_7$ | SPT$^1$ $\circlearrowright$    SSB $\to$ SYM $\to$ SPT$^4 \to$ SPT$^3 \to$ SPT$^6 \to$ SPT$^5 \to$ SPT$^2 \to$ SSB |

where the $x^{-1}$ should be understood as the modular multiplicative inverse of $x$ modulo $p$. Therefore, for any prime number $p$, there exists only one $p$-ality invariant SPT phase, which is given by,

$$a^2 - 2a + 1 = 0 \Rightarrow a = 1. \tag{127}$$

Then the $p$-ality P maps the gapped phases as,

$$\text{P}: \ \text{SPT}^1 \circlearrowright \quad \text{SSB} \to \text{SYM} \to \text{SPT}^{2^{-1}} \to \cdots \to \text{SPT}^2 \to \text{SSB}, \tag{128}$$

where the sequence of SPT phases follows (126) with the understanding that SPT$^0$ = SYM and "SPT$^\infty$" = SSB. For the first few prime numbers, the $p$-ality mapping is given in Tab. 3. We note that for $p = 2$ the $p$-ality coincides with the duality with off-diagonal bicharacter, i.e. Rep($D_8$), and for $p = 3$, the $p$-ality coincides with the triality Tri. The detailed mathematical structures and continuum field theory applications of $p$-ality are discussed in [56].

From this lattice perspective, we see that the $p$-ality always has a symmetric gapped phase, which is given by SPT$^1$. In general, if a theory is invariant under the $p$-ality, it admits the $p$-ality fusion category symmetry. We note that there are distinct $p$-ality fusion category symmetric gapped phases that cannot be continuously deformed to each other without breaking the non-invertible symmetry, similar to [24]. The $p$-ality non-invertible lines $\mathcal{P}_i, i = 1, \cdots, p-1$ have quantum dimension $p$ which follows the same argument as that in triality, namely, only the KW$^e$KW$^o$ part will contribute quantum dimension $p$, the other actions correspond to quantum dimension 1. The fusion rules for $\mathcal{P}_i$ are,

$$g \otimes \mathcal{P}_i = \mathcal{P}_i \otimes g = \mathcal{P}_i, \qquad \mathcal{P}_i \otimes \mathcal{P}_j = \begin{cases} \bigoplus_{g \in \mathbb{Z}_p \otimes \mathbb{Z}_p} g, & i = -j, \\ p \mathcal{P}_{i+j}, & \text{otherwise.} \end{cases} \tag{129}$$

## 7 Noninvertible symmetry from (un)twisted gauging

Given the partition function of the continuum field theory, the discrete gauging is specified by a bicharacter $\chi : G \times \widehat{G} \to U(1)$ and the possible discrete torsion $\alpha(g, h) = \varphi(g, h)\varphi(h, g)^{-1}$. We assume $G$ is abelian for simplicity. Suppose the original partition function has global symmetry $G$, which is tracked by the background gauge field $A$. We promote the background gauge field $A$ to the dynamical gauge field labeled by $a$. After gauging, there is a dual symmetry $\widehat{G}$ with background gauge field $\widehat{A}$,

$$\widetilde{Z}[X, \widehat{A}] = \frac{1}{|G|^g} \sum_{a \in H^1(X, G)} Z[X, a] \exp\left( 2\pi i \int \chi(a, \widehat{A}) + \alpha(a) \right), \tag{130}$$

where $X$ is the 2-dimensional base manifold, $a \in H^1(X, G)$ and $\widehat{A} \in H^1(X, \widehat{G})$. The bicharacter $\chi$ gives the identification between the original global symmetry and the dual symmetry. For

$\mathbb{Z}_2^{\text{e}} \times \mathbb{Z}_2^{\text{o}}$, if $\chi((A_1, A_2), (\widehat{A}_1, \widehat{A}_2)) = A_1 \cup \widehat{A}_1 + A_2 \cup \widehat{A}_2$ is diagonal pairing, then intuitively the dual symmetry is still $\mathbb{Z}_2^{\text{e}} \times \mathbb{Z}_2^{\text{o}}$. However, if $\chi((A_1, A_2), (\widehat{A}_1, \widehat{A}_2)) = A_1 \cup \widehat{A}_2 + A_2 \cup \widehat{A}_1$ is off-diagonal pairing, then the two $\mathbb{Z}_2$s get swapped. In particular,

$$\text{KW}^{\text{eo}} \sim \text{diagonal pairing} \sim \text{Rep}(H_8) = \text{TY}(\mathbb{Z}_2 \times \mathbb{Z}_2, \chi_{\text{diag}}, +1), \tag{131}$$

$$T\text{KW}^{\text{eo}} \sim \text{off-diagonal pairing} \sim \text{Rep}(D_8) = \text{TY}(\mathbb{Z}_2 \times \mathbb{Z}_2, \chi_{\text{offdiag}}, +1), \tag{132}$$

where $\text{KW}^{\text{eo}} = \text{KW}^{\text{e}} \circ \text{KW}^{\text{o}}$ as before and the last column gives the emerged infrared non-invertible symmetry if the theory is self-dual under gauging the $\mathbb{Z}_2 \times \mathbb{Z}_2$ symmetry with different bicharacters [61, 83, 84]. The translation $T$ will permute the even and odd sites, resulting in exchanging $\mathbb{Z}_2^{\text{e}}$ and $\mathbb{Z}_2^{\text{o}}$.

In general, if the theory is invariant under gauging an abelian symmetry $A$, then it admits the Tambara-Yamagami category symmetry $\text{TY}(A, \chi, \epsilon)$, where $\chi$ is the bicharater in the gauging, and $\epsilon \in H^3(\mathbb{Z}_2, U(1))$ is the Frobenius-Schur indicator. The simple objects in $\text{TY}(A, \chi, \epsilon)$ are group-like line operators $g \in A$ and a non-invertible line $\mathcal{N}$ with quantum dimension $\sqrt{|A|}$. These simple lines satisfy the following fusion rules,

$$g \otimes h = gh, \qquad g \otimes \mathcal{N} = \mathcal{N} \otimes g = \mathcal{N}, \qquad \mathcal{N} \otimes \mathcal{N} = \bigoplus_{g \in A} g. \tag{133}$$

The only non-trivial $F$-symbols are

$$[F_{\mathcal{N}}^{g\mathcal{N}h}]_{\mathcal{N},\mathcal{N}} = [F_h^{\mathcal{N}g\mathcal{N}}]_{\mathcal{N},\mathcal{N}} = \chi(g,h), \ [F_{\mathcal{N}}^{\mathcal{N}\mathcal{N}\mathcal{N}}]_{g,h} = \frac{\epsilon}{\sqrt{|A|}} \chi(g,h)^{-1}, \tag{134}$$

where $\epsilon = \pm 1$ is the Frobenius-Schur indicator for $\mathcal{N}$, which is classified by $\epsilon \in H^3(\mathbb{Z}_2, U(1)) = \mathbb{Z}_2$, and $\chi : A \times A \to U(1)$ is a non-degenerate symmetric bicharacter, which satisfies

$$\chi(g,h) = \chi(h,g), \qquad \chi(gh,k) = \chi(g,k)\chi(h,k), \qquad \chi(g,hk) = \chi(g,h)\chi(g,k). \tag{135}$$

For general $\mathbb{Z}_N \times \mathbb{Z}_N$, the untwisted gauging with diagonal and off-diagonal bicharacters are,

$$\text{KW}^{\text{eo}} : Z[A_1, A_2] \to Z[\widehat{A}_1, \widehat{A}_2] = \frac{1}{|G|^g} \sum_{a_1, a_2} Z[a_1, a_2] \omega^{a_1 \cup \widehat{A}_1 + a_2 \cup \widehat{A}_2}, \tag{136}$$

$$T\text{KW}^{\text{eo}} : Z[A_1, A_2] \to Z[\widehat{A}_1, \widehat{A}_2] = \frac{1}{|G|^g} \sum_{a_1, a_2} Z[a_1, a_2] \omega^{a_1 \cup \widehat{A}_2 + a_2 \cup \widehat{A}_1}. \tag{137}$$

The twisted gaugings $\text{TG}^1, \text{Tri} = TC^{\text{e}}\text{TG}^1$ and $\text{P} = TC^{\text{e}}\text{TG}^{-2}$ discussed in the previous section correspond to,

$$\text{TG}^1 = \text{KW}^{\text{eo}}\text{SPT} : Z[A_1, A_2] \to Z[\widehat{A}_1, \widehat{A}_2] = \frac{1}{|G|^g} \sum_{a_1, a_2} Z[a_1, a_2] \omega^{a_1 \cup a_2} \omega^{a_1 \cup \widehat{A}_1 + a_2 \cup \widehat{A}_2}, \tag{138}$$

$$\text{Tri} = TC^{\text{e}}\text{KW}^{\text{eo}}\text{SPT} : Z[A_1, A_2] \to Z[\widehat{A}_1, \widehat{A}_2] = \frac{1}{|G|^g} \sum_{a_1, a_2} Z[a_1, a_2] \omega^{a_1 \cup a_2} \omega^{a_1 \cup \widehat{A}_2 - a_2 \cup \widehat{A}_1}. \tag{139}$$

When $N = 2$, Tri transformation reduces to the triality in [61]. It is straightforward to verify $(\text{TG}^1)^3 : Z[A_1, A_2] \to Z[A_2, -A_1]$ and $(\text{Tri})^3 = \mathbb{1}$ using $a \cup b = -b \cup a$ and $\sum_a \omega^{a \cup b} = \delta_{b,0}$. One can also compute such partition function on a torus, the cup product becomes $A \cup B = A_x B_y - A_y B_x$, where $A_{x,y}$ denote the gauge fields along the cycles in the $x, y$-direction. The partition functions for the gapped phases are given by,

$$Z_{\text{SPT}^a}[A_1, A_2] = \omega^{aA_1 \cup A_2}, \qquad Z_{\text{SSB}}[A_1, A_2] = \delta_{A_1, 0} \delta_{A_2, 0}. \tag{140}$$

And the partially SSB phases are given by $\delta_{A,0}$, where $A$ is some combination of $A_1, A_2$ that corresponds to the diagonal subgroup. Lastly, the $p$-ality transformation $\mathsf{P} = T C^{\mathrm{e}} \mathsf{KW}^{\mathrm{eo}} \mathsf{SPT}^{-2}$ is give by,

$$\mathsf{P} : Z[A_1, A_2] \to Z[\widehat{A}_1, \widehat{A}_2] = \frac{1}{|G|^g} \sum_{a_1, a_2} Z[a_1, a_2] \omega^{-2a_1 \cup a_2} \omega^{a_1 \cup \widehat{A}_2 - a_2 \cup \widehat{A}_1}. \tag{141}$$

The better way to check the transformation on the boundary partition function is to examine the symmetry of bulk SymTFT as discussed in Appendix D.

For a theory that is invariant under the $\mathsf{Tri}$ or $\mathsf{P}$ transformation, it admits the triality or $p$-ality fusion category symmetry. Similar to the TY category, the pairing between the original symmetry and the dual symmetry relates to the $F$-symbols of the form of $F_{\mathcal{D}}^{g \mathcal{D} h}$, where $g, h \in \mathbb{Z}_N \times \mathbb{Z}_N$ and $\mathcal{D}$ is the triality or $p$-ality non-invertible defect. However, different from the TY category, the symmetry fractionalization class of triality or $p$-ality fusion category symmetry is in general not trivial, then, for example, $F_h^{\mathcal{D} \bar{\mathcal{D}} g}$ and $F_h^{g \mathcal{D} \bar{\mathcal{D}}}$ will be non-trivial [60, 99, 100].

## 7.1 Triality fusion category symmetry

If the $\mathbb{Z}_N \times \mathbb{Z}_N$ symmetric theory is invariant under the triality $\mathsf{Tri}$ transformation, then it admits the triality fusion category symmetry. Similar to the Tambara-Yamagami category, the triality fusion category can be viewed as an $\mathbb{Z}_3$ extension of $\mathrm{Vec}_{\mathbb{Z}_N \times \mathbb{Z}_N}$. The simple lines in the triality fusion category are group-like lines $g \in \mathbb{Z}_N \times \mathbb{Z}_N$ and non-invertible lines $\mathcal{Q}, \overline{\mathcal{Q}}$ with quantum dimension $N$,

$$g \otimes h = gh, \qquad g \otimes \mathcal{Q} = \mathcal{Q} \otimes g = \mathcal{Q}, \qquad g \otimes \overline{\mathcal{Q}} = \overline{\mathcal{Q}} \otimes g = \overline{\mathcal{Q}}, \tag{142}$$

$$\mathcal{Q} \otimes \mathcal{Q} = N \overline{\mathcal{Q}}, \qquad \mathcal{Q} \otimes \overline{\mathcal{Q}} = \bigoplus_{g \in \mathbb{Z}_N \times \mathbb{Z}_N} g, \tag{143}$$

which implies $\mathcal{Q} \otimes \mathcal{Q} \otimes \mathcal{Q} = N \bigoplus_{g \in \mathbb{Z}_N \times \mathbb{Z}_N} g$. In the following, we try to determine the triality fusion category that is realized in the $\mathbb{Z}_N \times \mathbb{Z}_N$ spin model. The triality fusion category of this particular twisted gauging $\mathsf{Tri}$ with corresponding transformation on the partition function(139) can be a group theoretical fusion category,

$$\mathcal{C}((\mathbb{Z}_N^a \times \mathbb{Z}_N^b) \rtimes \mathbb{Z}_3, \omega_\kappa, \mathbb{Z}_N^a, 1), \tag{144}$$

where

$$(\mathbb{Z}_N^a \times \mathbb{Z}_N^b) \rtimes \mathbb{Z}_3 = \langle a, b, c | a^N = b^N = c^3 = 1, ab = ba, cac^{-1} = a^{-1}b, cbc^{-1} = a^{-1} \rangle, \tag{145}$$

where $\mathbb{Z}_3$ is the subgroup of the automorphism group of $\mathbb{Z}_N^a \times \mathbb{Z}_N^b$, $\mathrm{Aut}(\mathbb{Z}_N^a \times \mathbb{Z}_N^b) = \mathrm{GL}(2, \mathbb{Z}_N)$ assuming $N$ is prime. $\omega_\kappa$ relates to the Frobenius-Schur (FS) indicator of $\mathcal{Q}$, and its explicit form is given in [56]. For this triality fusion category, the non-trivial FS indicator forbids a symmetric gapped phase. Therefore, in the case that has $\mathsf{Tri}$ invariant gapped phase, the triality fusion category is necessarily anomaly-free with a trivial FS indicator. Hence, for $N = 3$ and prime $N = 1 \mod 3$, the triality fusion category is $\mathcal{C}((\mathbb{Z}_N^a \times \mathbb{Z}_N^b) \rtimes \mathbb{Z}_3, 1, \mathbb{Z}_N^a, 1)$.

The group theoretical fusion category suggests that the triality transformation can be obtained by gauging the subgroup $\mathbb{Z}_N^a$ of $(\mathbb{Z}_N^a \times \mathbb{Z}_N^b) \rtimes \mathbb{Z}_3$. One starts with the theory that is symmetric under $(\mathbb{Z}_N^a \times \mathbb{Z}_N^b) \rtimes \mathbb{Z}_3$, then gauges the $\mathbb{Z}_N^a$ subgroup, the $\mathbb{Z}_3$ symmetry transformation becomes triality transformation as illustrated in [56].

In particular, we consider the $\mathbb{Z}_N^{\mathrm{e}} \times \mathbb{Z}_N^{\mathrm{o}}$ spin model, there are 3 particular partially spontaneously symmetry breaking phases,

$$H_{\mathrm{SSB}^{\mathrm{e}}} = -\sum_j Z_{2j} Z_{2j+2}^{-1} + X_{2j+1} + h.c., \tag{146}$$

$$H_{\mathrm{SSB}^{\mathrm{o}}} = -\sum_j Z_{2j-1} Z_{2j+1}^{-1} + X_{2j} + h.c., \tag{147}$$

$$H_{\mathrm{SSB}^{\mathrm{diag}}} = -\sum_j Z_{2j-2} Z_{2j-1}^{-1} Z_{2j}^{-1} Z_{2j+1} + X_{2j-2} X_{2j-1} + h.c. \tag{148}$$

The $\mathbb{Z}_3$ invertible symmetry permutes these 3 partially SSB phases, while it becomes triality Tri non-invertible symmetry under the conjugation of $\mathsf{KW}^{\mathrm{e}}$ or gauging $\mathbb{Z}_N^{\mathrm{e}}$,

$$
\begin{array}{ccc}
& \mathrm{SSB}^{\mathrm{diag}} & & & \mathrm{SPT}^{-1} \\
\mathrm{SSB}^{\mathrm{e}} \overset{\nearrow\quad\searrow}{\underset{\longleftarrow}{\phantom{xxx}}} \mathrm{SSB}^{\mathrm{o}} & \overset{\mathsf{KW}^{\mathrm{e}}}{\longleftrightarrow} & \mathrm{SYM} \overset{\nearrow\quad\searrow}{\underset{\longleftarrow}{\phantom{xxx}}} \mathrm{SSB}.
\end{array}
\tag{149}
$$

In partition function formalism, the $\mathbb{Z}_3$ symmetry acts as,

$$\mathbb{Z}_3: \quad Z[A_1, A_2] \rightarrow Z[-A_1 + A_2, -A_1], \tag{150}$$

where $A_1, A_2$ are the background gauge fields of $\mathbb{Z}_N^{\mathrm{e}}, \mathbb{Z}_N^{\mathrm{o}}$. We define the $\mathbb{Z}_N^{\mathrm{e}}$ gauged partition function,

$$\widehat{Z}[A_1, A_2] = \frac{1}{|G|^g} \sum_{a_1} Z[a_1, A_2] \omega^{a_1 \cup A_1}. \tag{151}$$

Then the $\mathbb{Z}_3$ acts on the gauged theory as,

$$\widehat{\mathbb{Z}}_3: \quad \widehat{Z}[A_1, A_2] = \frac{1}{|G|^g} \sum_{a_1} Z[a_1, A_2] \omega^{a_1 \cup A_1} \tag{152}$$

$$\longrightarrow \frac{1}{|G|^g} \sum_{a_1} Z[-a_1 + A_2, -a_1] \omega^{a_1 \cup A_1} \tag{153}$$

$$= \frac{1}{|G|^g} \sum_{a_1, a_2} \widehat{Z}[a_2, a_1] \omega^{-a_1 \cup A_1 + a_2 \cup a_1 + a_2 \cup A_2} \tag{154}$$

$$= \frac{1}{|G|^g} \sum_{a_1, a_2} \widehat{Z}[a_1, a_2] \omega^{a_1 \cup a_2} \omega^{a_1 \cup A_2 - a_2 \cup A_1}, \tag{155}$$

which agrees with the transformation in (139).

## 7.2 $p$-ality fusion category symmetry

In the following discussion, $p$ is a prime number. If the $\mathbb{Z}_p \times \mathbb{Z}_p$ symmetric theory is further invariant under the $p$-ality $\mathsf{P}$ transformation, then it admits the $p$-ality fusion category symmetry. We then try to determine the $p$-ality fusion category that is realized in the $\mathbb{Z}_p \times \mathbb{Z}_p$ spin models. The fusion rules of the simple lines are given by,

$$g \otimes h = gh, \qquad g \otimes \mathcal{P}_i = \mathcal{P}_i \otimes g = \mathcal{P}_i, \qquad \mathcal{P}_i \otimes \mathcal{P}_j = \begin{cases} \bigoplus_{g \in \mathbb{Z}_p \otimes \mathbb{Z}_p} g, & i = -j, \\ p\mathcal{P}_{i+j}, & \text{otherwise.} \end{cases} \tag{156}$$

The complete classification of the $p$-ality fusion category is not known. However, one particularly interesting one is the group theoretical fusion category [56],

$$\mathcal{P}_{+,m} = \mathcal{C}(\mathbb{Z}_p^a \times \mathbb{Z}_p^b \times \mathbb{Z}_p^c, \omega_{+,m}, \mathbb{Z}_p^a \times \mathbb{Z}_p^b, 1), \tag{157}$$

where $\omega_{+,m}(a^{i_1} b^{j_1} c^{k_1}, a^{i_2} b^{j_2} c^{k_2}, a^{i_3} b^{j_3} c^{k_3}) = e^{\frac{2\pi i}{p} i_1 j_2 k_3 + \frac{2\pi i m}{p^2} k_1(k_2 + k_3 - [k_2 + k_3]_p)}$. The second part in $\omega_{+,m}$ is a type I anomaly of $\mathbb{Z}_p^c$ and it relates to Frobenius-Schur indicator of $\mathcal{P}_i$. In this $p$-ality fusion category, the non-trivial FS indicator obstructs the symmetric trivial gapped phase. Our interested models have $\text{SPT}^1$ as the $p$-ality invariant theory, therefore, the FS indicator must be trivial. To realize non-trivial FS indicators, one needs to stack SPT corresponding to the non-trivial element in $H^3(\mathbb{Z}_p, U(1))$ [21].

The group theoretical fusion category suggests the $p$-ality is obtained by gauging $\mathbb{Z}_p^a \times \mathbb{Z}_p^b$ subgroup of global symmetry $\mathbb{Z}_p^a \times \mathbb{Z}_p^b \times \mathbb{Z}_p^c$ with type III anomaly. The type III anomaly is used to bootstrap the conformal field theory of the multicritical point between the SPT phases [58]. For $p = 2$, this analysis coincides with study of $\text{Rep}(D_8) \cong \mathcal{C}(\mathbb{Z}_2 \times \mathbb{Z}_2 \times \mathbb{Z}_2, \omega_{+,0}, \mathbb{Z}_2 \times \mathbb{Z}_2, 1)$ [101]. In particular, [24] uses the Kennedy-Tasaki transformation to convert the $\text{Rep}(D_8)$ to $\mathbb{Z}_2 \times \mathbb{Z}_2 \times \mathbb{Z}_2$ with the type III anomaly, and then analyze the different symmetry breaking patterns to construct the symmetric gapped phases (fiber functors) of $\text{Rep}(D_8)$. Parallel analysis can yield other symmetric gapped phases of this $p$-ality fusion category.

To be specific, we consider the $\mathbb{Z}_p^e \times \mathbb{Z}_p^o$ spin system as in the previous section, where the symmetry is generated by $\eta^e = \prod_j X_{2j}, \eta^o = \prod_j X_{2j-1}$. The $\mathbb{Z}_p^c$ symmetry is generated by

$$\eta^c = \prod_j \text{CZ}_{2j-1,2j}^{-1} \text{CZ}_{2j,2j+1}^1, \tag{158}$$

which permutes the $\text{SPT}^a \to \text{SPT}^{a+1}$ but leaves the SSB phase invariant,

$$\mathbb{Z}_p^c: \quad \text{SSB} \circlearrowleft \quad \text{SYM} \to \text{SPT}^1 \to \text{SPT}^2 \to \cdots \to \text{SPT}^{p-1} \to \text{SYM}. \tag{159}$$

Following the group theoretical fusion category, the $p$-ality fusion category is obtained by gauging $\mathbb{Z}_p^e \times \mathbb{Z}_p^o$. Note that stacking $\mathbb{Z}_p^e \times \mathbb{Z}_p^o$ SPTs will not change the fusion category since the type III anomaly will assimilate the SPT. In the following, we consider the $\text{TS}_1\text{S}_2\text{T}^{-1}$ transformation, which is the $\mathbb{Z}_p$ generalization of Kennedy-Tasaki transformation, $\text{SPT KW}^{e,o} \text{SPT}^{-1}$. Note that this is an order 2 non-local mapping,

$$\text{SSB} \xleftrightarrow{\text{TST}^{-1}} \text{SPT}^1, \qquad \text{SPT}^a \xleftrightarrow{\text{TST}^{-1}} \text{SPT}^{1-(a-1)^{-1}}, \tag{160}$$

where the inverse should be understood as $\mod p$. This matches with the (128),

$$\text{P}: \quad \text{SPT}^1 \circlearrowleft \quad \text{SPT}^2 \to \text{SSB} \to \text{SYM} \to \cdots \to \text{SPT}^{3 \cdot 2^{-1}} \to \text{SPT}^2. \tag{161}$$

We follow the procedure in the previous section and [56] to derive the $p$-ality action on the partition function. Because of the type III anomaly, the $\mathbb{Z}_p^c$ symmetry acts on the partition function as,

$$\mathbb{Z}_p^c: \quad Z[A_1, A_2] \to Z[A_1, A_2] \omega^{A_1 \cup A_2}, \tag{162}$$

where $A_1, A_2$ tracks the global symmetry $\mathbb{Z}_p^e$ and $\mathbb{Z}_p^o$. We define the $\mathbb{Z}_p^e \times \mathbb{Z}_p^o$ gauged partition function following the $\text{TS}_1\text{S}_2\text{T}^{-1}$ transformation,

$$\widehat{Z}[A_1, A_2] = \frac{1}{|G|^g} \sum_{a_1, a_2} Z[a_1, a_2] \omega^{-a_1 \cup a_2 + a_1 \cup A_1 + a_2 \cup A_2 + A_1 \cup A_2}. \tag{163}$$

Then the $\mathbb{Z}_p^c$ symmetry acts on the gauged theory as,

$$\widehat{\mathbb{Z}}_p^c : \widehat{Z}[A_1, A_2] = \frac{1}{|G|^g} \sum_{a_1, a_2} Z[a_1, a_2] \omega^{-a_1 \cup a_2 + a_1 \cup A_1 + a_2 \cup A_2 + A_1 \cup A_2} \tag{164}$$

$$\rightarrow \frac{1}{|G|^g} \sum_{a_1, a_2} Z[a_1, a_2] \omega^{a_1 \cup a_2} \omega^{-a_1 \cup a_2 + a_1 \cup A_1 + a_2 \cup A_2 + A_1 \cup A_2}$$

$$= \frac{1}{|G|^g} \sum_{a_1, a_2; b_1, b_2} \widehat{Z}[b_1, b_2] \omega^{-b_1 \cup b_2 + b_1 \cup a_1 + b_2 \cup a_2 + a_1 \cup a_2} \omega^{a_1 \cup A_1 + a_2 \cup A_2 + A_1 \cup A_2} \tag{165}$$

$$= \frac{1}{|G|^g} \sum_{b_1, b_2} \widehat{Z}[b_1, b_2] \omega^{-2b_1 \cup b_2 + b_1 \cup A_2 - b_2 \cup A_1} . \tag{166}$$

This matches with (141).

Furthermore, another group theoretical fusion category $\mathcal{P}_{-,m}$ in [56] involves type II anomaly, and it is also compatible with the fusion rules in (156). In this case, the non-trivial FS indicator can still have a symmetric trivial gapped phase, since the type II anomaly will "cancel" the obstruction. It is also interesting to construct various symmetric gapped phases of this $p$-ality fusion category $\mathcal{P}_{-,m}$.

# 8 Conclusion

In this paper, we studied the (un)twisted gauging on a 1d lattice incorporating the analogous data from continuous theory. The non-local duality mapping is translated into the gauging procedure, and the general (un)twisted gauging leads to novel non-local mapping of the local Hamiltonians, which preserves the locality of symmetric operators but maps charged operators to non-local operators. The dual symmetry is explicit in this construction and could be fusion category symmetry under the (un)twisted gauging. We elaborate the (un)twisted gauging on lattice using both lattice operator algebra and Kraus operator in the quantum process. We detailed study the triality for $\mathbb{Z}_N \times \mathbb{Z}_N$ symmetric Hamiltonians and $p$-ality mapping for the $\mathbb{Z}_p \times \mathbb{Z}_p$ symmetric Hamiltonians with prime $p$, we found the mapping among different gapped phases. Under certain conditions, there exist triality or $p$-ality invariant gapped phases. For theories that are invariant under the triality or $p$-ality, they admit corresponding non-invertible symmetry. We analyze their corresponding group theoretical fusion category construction. The data of the non-invertible symmetry contains additional information of the triality or $p$-ality defects, which can be extracted from the lattice defect Hamiltonians as in [21]. For the non-invertible symmetric gapped phases with unique ground state, there are distinguishable cousins which cannot be smoothly connected to each other.

There are several directions to pursue in the future:

- Since we know the triality and $p$-ality fusion category admit symmetric gapped phases with unique ground state under certain algebraic condition, it is interesting to have the classification of these symmetric gapped phases. For group theoretical fusion category, the condition is known in [56,59,60]. The lattice construction of these trivial symmetric gapped phases is leaving for future study. One approach is to import the fusion category data to the tensor network formalism as developed in [120].

- It would be interesting to generalize the lattice twisted gauging as an ingredient to compare and generate other interesting non-local mapping with respect to non-abelian symmetry, fermionic symmetry, non-invertible symmetry and higher form symmetry. Many related works on duality mappings for various cases are reviewed in the introduction.

- It is also interesting to extract more fusion category data and the data of fiber functors from the lattice (defect) Hamiltonian [21, 24, 61, 93, 143].

- The (1+1)d symmetric gapped phases with domains can be used to encode the logical qubits and using non-local mapping as the quantum gate to implement the quantum computing. This corresponds to the boundary perspective of the dynamic automorphism codes [144–148].

## Acknowledgments

We acknowledge the discussions with Yimu Bao, Yichul Choi, Sheng-Jie Huang, Laurens Lootens, Chong Wang, Xinping Yang, Zipei Zhang, Liujun Zou. We acknowledge Zipei Zhang for the collaboration on a related project. We thank Sahand Seifnashri, Shu-Heng Shao for helpful comments on the draft.

**Funding information**   DCL and YZY are supported by a startup fund by UCSD and the National Science Foundation (NSF) Grant No. DMR-2238360. This research was supported in part by grant NSF PHY-2309135 to the Kavli Institute for Theoretical Physics (KITP), through the KITP Program "Correlated Gapless Quantum Matter" (2024).

## A   Twisted Gauss law operator

For $\mathbb{Z}_N \times \mathbb{Z}_N$, the non-trivial element in $H^2(\mathbb{Z}_N \times \mathbb{Z}_N, U(1))$ is $\varphi(g, h) = \omega^{b g_1 h_2}$, where $b \in \mathbb{Z}_N$. The general Gauss law operator is given by,

$$
\begin{aligned}
G_j^{(g_1, g_2)} &= \sum_{a, b \in G} \left( \varphi(ag^{-1}, g)^\dagger \left| ag^{-1} \right\rangle \langle a | \right) \otimes \lambda_j^g \otimes \left( \varphi(g, b) | gb \rangle \langle b | \right) \\
&= \sum_{(a_1, a_2), (b_1, b_2)} \left( \omega^{-b(a_1 - g_1) g_2} | a_1 - g_1 \rangle \langle a_1 | \right)_{j - \frac{1}{2}} \otimes \left( | a_2 - g_2 \rangle \langle a_2 | \right)_{j - \frac{1}{2}} \otimes (X_j^{\mathrm{e}})^{g_1} (X_j^{\mathrm{o}})^{g_2} \\
&\quad \otimes \left( | g_1 + b_1 \rangle \langle b_1 | \right)_{j + \frac{1}{2}} \otimes \left( \omega^{b g_1 b_2} | g_2 + b_2 \rangle \langle b_2 | \right)_{j + \frac{1}{2}} \\
&= (\widetilde{Z}_{j - \frac{1}{2}}^{\mathrm{e}})^{-b g_2} (\widetilde{X}_{j - \frac{1}{2}}^{\mathrm{e}})^{g_1} (\widetilde{X}_{j - \frac{1}{2}}^{\mathrm{o}})^{g_2} (X_j^{\mathrm{e}})^{g_1} (X_j^{\mathrm{o}})^{g_2} (\widetilde{X}_{j + \frac{1}{2}}^{\mathrm{e}})^{-g_1} (\widetilde{X}_{j + \frac{1}{2}}^{\mathrm{o}})^{-g_2} (\widetilde{Z}_{j + \frac{1}{2}}^{\mathrm{o}})^{b g_1},
\end{aligned}
\tag{A.1}
$$

where $(a_1, a_2), (b_1, b_2), (g_1, g_2) \in \mathbb{Z}_N^{\mathrm{e}} \times \mathbb{Z}_N^{\mathrm{o}}$ and $|a\rangle_{j - \frac{1}{2}} = (|a_1\rangle \otimes |a_2\rangle)_{j - \frac{1}{2}}$. Let $(g_1, g_2) = (1, 0)$ and $(0, 1)$, we have,

$$
G_j^{(1,0)} = \widetilde{X}_{j - \frac{1}{2}}^{\mathrm{e}} (X_j^{\mathrm{e}})^{g_1} (\widetilde{X}_{j + \frac{1}{2}}^{\mathrm{e}})^{-1} (\widetilde{Z}_{j + \frac{1}{2}}^{\mathrm{o}})^b, \qquad G_j^{(0,1)} = (\widetilde{Z}^{\mathrm{e}})_{j - \frac{1}{2}}^{-b} \widetilde{X}_{j - \frac{1}{2}}^{\mathrm{o}} X_j^{\mathrm{o}} (\widetilde{X}_{j - \frac{1}{2}}^{\mathrm{o}})^{-1}.
\tag{A.2}
$$

## B   Useful identities for $\mathbb{Z}_N$ generalized Pauli matrices

As defined in the main text,

$$
C\mathcal{O}_{m,n} = \frac{1}{N} \sum_{\alpha=1}^{N} \sum_{\beta=1}^{N} \omega^{-\alpha\beta} Z_m^\alpha \mathcal{O}_n^\beta.
\tag{B.1}
$$

Under the $CZ^a$ transformation,

$$
XI \to XZ^a, \qquad IX \to Z^a X, \qquad ZI \to ZI, \qquad IZ \to IZ,
\tag{B.2}
$$

whose symplectic transformation is,

$$T_{\mathsf{CZ}^a} = \begin{pmatrix} 1 & 0 & 0 & 0 \\ 0 & 1 & 0 & 0 \\ 0 & a & 1 & 0 \\ a & 0 & 0 & 1 \end{pmatrix}. \tag{B.3}$$

Under the $\mathsf{CX}^a$ transformation,

$$XI \to XX^a, \qquad IX \to IX, \qquad ZI \to ZI, \qquad IZ \to Z^{-a}Z, \tag{B.4}$$

whose symplectic transformation is,

$$T_{\mathsf{CX}^a} = \begin{pmatrix} 1 & 0 & 0 & 0 \\ a & 1 & 0 & 0 \\ 0 & 0 & 1 & -a \\ 0 & 0 & 0 & 1 \end{pmatrix}. \tag{B.5}$$

We can also define the generalized Hadamard gate and its corresponding symplectic transformation,

$$\mathsf{H} = \frac{1}{\sqrt{N}} \omega^{-(i-1)(j-1)}, \qquad T_{\mathsf{H}} = \begin{pmatrix} 0 & 1 \\ -1 & 0 \end{pmatrix}. \tag{B.6}$$

# C Practical review of algebraic methods for quantum codes on lattice

In this section, we give a practical review of the algebraic methods for quantum codes on lattice developed in [79,80] and the nice review [81]. The basic idea is that the Pauli operators can be mapped to vectors in the symplectic vector spaces to keep the commutation relation but forget the phases in front of the Pauli operators. The elements in the Clifford group, which maps any Pauli operator to a Pauli operator, are represented as matrices that preserve the symplectic structure.

First, let's consider a quNit, with local Hilbert space $\mathbb{C}^N$, spanned by $|n\rangle$, $n \in \mathbb{Z}_N$. We define the generalized Pauli operator as in the main context, $X|n\rangle = |n-1\rangle$, $Z|n\rangle = \omega^n |n\rangle$, where $\omega = e^{i\frac{2\pi}{N}}$. And $XZ = \omega ZX$. The general Pauli operator is given by $\eta X^\xi Z^\zeta$, where $\xi, \zeta \in \mathbb{Z}_N$ and $\eta$ is the phase factor. Two Pauli operators have the commutation relation,

$$\eta X^\xi Z^\zeta \eta' X^{\xi'} Z^{\zeta'} = \omega^{\xi\zeta' - \xi'\zeta} \eta' X^{\xi'} Z^{\zeta'} \eta X^\xi Z^\zeta. \tag{C.1}$$

And they commute up to a phase $\omega^m$, where

$$m = \xi\zeta' - \xi'\zeta \mod N = \begin{pmatrix} \xi & \zeta \end{pmatrix} \begin{pmatrix} 0 & 1 \\ -1 & 0 \end{pmatrix} \begin{pmatrix} \xi' \\ \zeta' \end{pmatrix} \mod N. \tag{C.2}$$

The Pauli operator $\eta X^\xi Z^\zeta$ is represented by the vector $(\xi\ \zeta)^{\mathsf{T}}$. And the commutation relation between two Pauli operators is encoded by,

$$\begin{pmatrix} \xi & \zeta \end{pmatrix} \begin{pmatrix} 0 & 1 \\ -1 & 0 \end{pmatrix} \begin{pmatrix} \xi' \\ \zeta' \end{pmatrix} \mod N. \tag{C.3}$$

Any transformation that preserves the commutation relation is a symplectic transformation on the vector, for the symplectic transformation $A$ is a $2 \times 2$ matrix with values in $\mathbb{Z}_N$,

$$A^{\mathsf{T}} \begin{pmatrix} 0 & 1 \\ -1 & 0 \end{pmatrix} A = \begin{pmatrix} 0 & 1 \\ -1 & 0 \end{pmatrix}. \tag{C.4}$$

For example, the elementary transformations are,

$$\begin{pmatrix} 1 & 0 \\ a & 1 \end{pmatrix}, \qquad \begin{pmatrix} 0 & 1 \\ -1 & 0 \end{pmatrix}, \qquad \begin{pmatrix} a & 0 \\ 0 & a^{-1} \end{pmatrix}, \tag{C.5}$$

where $a \in \mathbb{Z}_N^\times$. They correspond to the phase gate, Hadamard gate, and the $K_d = \sum_i |-i\rangle \langle i|$ gate [149]. For $m$ quNit system, the Pauli operator is represented as $2m$-vector,

$$\prod_{l=1}^m (X^l)^{\xi^l}(Z^l)^{\zeta^l} \rightsquigarrow \begin{pmatrix} \xi^1 & \cdots & \xi^m & | & \zeta^1 & \cdots & \zeta^m \end{pmatrix}^\intercal, \tag{C.6}$$

where $X^l$ acts on the $l$-th quNit, similar for $Z^l$. We neglect the phase factors in front of the Pauli operators since they are irrelevant for the commutation relation. Then the symplectic transformation is given by $2m \times 2m$ matrix with values in $\mathbb{Z}_N$ and satisfies,

$$A^\intercal \begin{pmatrix} 0 & I \\ -I & 0 \end{pmatrix} A = \begin{pmatrix} 0 & I \\ -I & 0 \end{pmatrix}, \qquad A \in \mathrm{GL}(2m, \mathbb{Z}_N), \tag{C.7}$$

where $I$ is $m \times m$ identity matrix. The symplectic transformations correspond to CZ and CX are listed in Appendix B.

It is interesting and useful to incorporate the translation invariance in the algebraic method. In one spatial dimension, the translation symmetry is $\mathbb{Z}$ and generated by $x$. $x^i$ represents translation by $i$ sites (unit cells). Then the Pauli operator $\prod_i X_i^{\xi_i} Z_i^{\zeta_i}$, where $X_i, Z_i$ act on the site (unit cell) $i$, is represented as,

$$\begin{pmatrix} \sum_i \xi_i x^i & | & \sum_i \zeta_i x^i \end{pmatrix}^\intercal. \tag{C.8}$$

And the dual vector is given by $x \to x^{-1}$,

$$\begin{pmatrix} \sum_i \xi_i x^{-i} & | & \sum_i \zeta_i x^{-i} \end{pmatrix}. \tag{C.9}$$

The commutation relation between two Pauli operators is encoded by,

$$\mathrm{xtr}\left[ \begin{pmatrix} \sum_i \xi_i x^{-i} & \sum_i \zeta_i x^{-i} \end{pmatrix} \begin{pmatrix} 0 & 1 \\ -1 & 0 \end{pmatrix} \begin{pmatrix} \sum_i \xi_i' x^i \\ \sum_i \zeta_i' x^i \end{pmatrix} \right], \tag{C.10}$$

where $\mathrm{xtr}[x^i] = 1$ if $i = 0$ and $0$ otherwise. For example, the Ising coupling $Z_i Z_{i+1}^{-1}$ is represented as,

$$\begin{pmatrix} 0 & | & x^i - x^{i+1} \end{pmatrix}^\intercal \simeq \begin{pmatrix} 0 & | & 1 - x \end{pmatrix}^\intercal. \tag{C.11}$$

The $\simeq$ is because of the translation invariance. For $m$ quNit system on 1-dimensional lattice with translation invariance, the general Pauli operator is given by $\prod_{l=1}^m \prod_i (X_i^l)^{\xi_i^l}(Z_i^l)^{\zeta_i^l}$, where $X_i^l$ acts on $l$-th quNit and site $i$. It is represented as,

$$\prod_{l=1}^m \prod_i (X_i^l)^{\xi_i^l}(Z_i^l)^{\zeta_i^l} \rightsquigarrow \sum_i x^i \begin{pmatrix} \xi_i^1 & \cdots & \xi_i^m & | & \zeta_i^1 & \cdots & \zeta_i^m \end{pmatrix}^\intercal. \tag{C.12}$$

Any transformation $A(x)$ that preserves the commutation relation is given by,

$$A(x^{-1})^\intercal \begin{pmatrix} 0 & I \\ -I & 0 \end{pmatrix} A(x) = \begin{pmatrix} 0 & I \\ -I & 0 \end{pmatrix}, \tag{C.13}$$

where $A(x)$ is a polynomial of $x$ with coefficients as $2m \times 2m$ matrices with values in $\mathbb{Z}_N$. For example, on the 1d lattice with one quNit per site, some interesting unitary is given by,

$$\prod_j \mathrm{CZ}_{j,j+1} \rightsquigarrow \begin{pmatrix} 1 & 0 \\ x^{-1} + x & 1 \end{pmatrix}, \qquad \prod_j \mathrm{CX}_{j,j+1} \to \begin{pmatrix} \frac{1}{1-x} & 0 \\ 0 & -x^{-1} + 1 \end{pmatrix}, \tag{C.14}$$

where the second one is acting $\mathrm{CX}_{j,j+1}$ sequentially and it does not preserve locality. For local Hilbert space with $m$ quNit, some elementary symplectic transformations are given by,

- Hadamard $\widetilde{\mathsf{H}}_i$: $\text{row}_i = \text{row}_j$, $\text{row}_j = -\text{row}_i$ for $1 \le i \le m$,

- controlled-NOT $\widetilde{\mathsf{C}}_{i\to j}(a)$: $\text{row}_i \mathrel{+}= a(x) \times \text{row}_j$, $\text{row}_{j+m} \mathrel{+}= -a(x^{-1}) \times \text{row}_{i+m}$ for $1 \le i \ne j \le m$,

- controlled-Phase: $\text{row}_{i+m} \mathrel{+}= f \times \text{row}_i$ where $f \in \mathbb{Z}_N$ for $1 \le i \le m$,

where $a(x)$ is a polynomial of $x$ with coefficients in $\mathbb{Z}_N$.

For example, the stabilizers for the $\mathbb{Z}_N \times \mathbb{Z}_N$ symmetric phase (1,2-column), SSB phase (3,4-column) and SPT phase (5,6-column) with minimal coupling to the $\mathbb{Z}_N \times \mathbb{Z}_N$ gauge field is represented as

$$
\begin{pmatrix}
\text{SYM} & & \text{SSB} & & \text{SPT} & & \text{Gauss law} & \\
0 & 1 & 0 & 0 & x & 0 & 0 & x \\
0 & 0 & 0 & 0 & 0 & 0 & 0 & 1-x \\
1 & 0 & 0 & 0 & 0 & 1 & x & 0 \\
0 & 0 & 0 & 0 & 0 & 0 & 1-x & 0 \\
0 & 0 & 0 & 1-x & 0 & ax-a & 0 & 0 \\
0 & 0 & 0 & 1 & 0 & -a & bx & 0 \\
0 & 0 & 1-x & 0 & a-ax & 0 & 0 & 0 \\
0 & 0 & 1 & 0 & a & 0 & 0 & -b
\end{pmatrix},
\tag{C.15}
$$

where the basis is $(X_{2j-1}, \widetilde{X}_{2j-\frac{1}{2}}, X_{2j}, \widetilde{X}_{2j+\frac{1}{2}}, Z_{2j-1}, \widetilde{Z}_{2j-\frac{1}{2}}, Z_{2j}, \widetilde{Z}_{2j+\frac{1}{2}})$ and the last two columns correspond to the twisted Gauss law operators (96). To find the unitary that transforms the twisted Gauss law operator (96) into the form of single $X$, one applies the elementary symplectic transformations to (C.15),

$$
\widetilde{\mathsf{H}}_4 \circ \widetilde{\mathsf{H}}_1 \circ \widetilde{\mathsf{C}}_{4\to 1}(bx) \circ \widetilde{\mathsf{C}}_{2\to 1}(x) \circ \widetilde{\mathsf{C}}_{2\to 1}(-1) \circ \widetilde{\mathsf{C}}_{3\to 2}(-b) \circ \widetilde{\mathsf{C}}_{3\to 4}(-x^{-1}) \circ \widetilde{\mathsf{C}}_{3\to 4}(1) \circ \widetilde{\mathsf{H}}_1 \circ \widetilde{\mathsf{H}}_2, \tag{C.16}
$$

which corresponds to the matrix form in (101). Certainly, there are different paths lead to the same unitary transformation. It is an interesting question to find the minimal one. Under the unitary transformation, the resulting stabilizers are,

$$
\begin{pmatrix}
0 & -1 & 0 & 0 & -x & 0 & 0 & -x \\
-b & 0 & 0 & 1 & 0 & -a-b & 0 & 0 \\
1 & 0 & 0 & 0 & 0 & 1 & x & 0 \\
0 & \frac{b}{x} & 1 & 0 & a+b & 0 & 0 & 0 \\
0 & 0 & 0 & 0 & 0 & 0 & 0 & 0 \\
0 & \frac{1}{x}-1 & 0 & 0 & 1-x & 0 & 0 & 0 \\
0 & 0 & 0 & 0 & 0 & 0 & 0 & 0 \\
\frac{1}{x}-1 & 0 & 0 & 0 & 0 & \frac{1}{x}-1 & 0 & 0
\end{pmatrix}.
\tag{C.17}
$$

# D  Bulk symmetry TFT

The $\mathbb{Z}_N \times \mathbb{Z}_N$ symmetric theory in (1+1)d can be viewed as the boundary of (2+1)d $\mathbb{Z}_N \times \mathbb{Z}_N$ quantum double. Using the Chern-Simons theory representation,

$$
\mathcal{L} = \frac{1}{4\pi} K_{IJ} a^I \wedge da^J, \qquad K = N \begin{pmatrix} 0 & 0 & 1 & 0 \\ 0 & 0 & 0 & 1 \\ 1 & 0 & 0 & 0 \\ 0 & 1 & 0 & 0 \end{pmatrix},
\tag{D.1}
$$

the charge vector is $(e_1, e_2, m_1, m_2)^{\intercal}$. The above Chern-Simons theory is a continuum description of the $\mathbb{Z}_N \times \mathbb{Z}_N$ Dijkgraaf-Witten theory [150–152]. The non-local mappings of the boundary theories correspond to the global symmetry of the bulk theory [153, 154]. In particular, the bulk global symmetry of $\mathbb{Z}_N^m$ quantum double is generated by $V$,

$$\left\{ V \in \mathrm{GL}(2m, \mathbb{Z}_N) \mid V \begin{pmatrix} 0 & I \\ I & 0 \end{pmatrix} V^{\intercal} = \begin{pmatrix} 0 & I \\ I & 0 \end{pmatrix} \right\}, \tag{D.2}$$

where $I$ is $m \times m$ identity matrix. Such a group is called a split orthogonal group. There are 3 types of bulk global symmetry, and they correspond to different actions on the boundary theory [153].

**R-type**  The R-type symmetry in the matrix form is,

$$\mathrm{R}(U) = \begin{pmatrix} U & 0 \\ 0 & U^{-1\intercal} \end{pmatrix}, \qquad A^{\intercal} \to U^{-1\intercal} A^{\intercal}, \tag{D.3}$$

where $A$ is the background gauge fields in the boundary theory that track the global symmetry. Note that such bulk global symmetry corresponds to the automorphism of the boundary symmetry group.

**T-type**  The T-type symmetry is generated by,

$$\mathrm{T}^n = \begin{pmatrix} I & \begin{pmatrix} 0 & n \\ -n & 0 \end{pmatrix} \\ 0 & I \end{pmatrix}, \qquad Z[A_1, A_2] \to Z[A_1, A_2] \omega^{n A_1 \cup A_2}, \tag{D.4}$$

where $I$ is the identity matrix and $Z[A_1, A_2]$ is the partition function of the boundary theory. The T-type bulk symmetry corresponds to applying an SPT entangler on the boundary theory.

**S-type**  The S-type symmetry is given by,

$$\mathrm{S}_i = \begin{pmatrix} I - J & J \\ J & I - J \end{pmatrix}, \qquad Z[\cdots, A_i, \cdots] \to \sum_{a_i \in H^1(X, \mathbb{Z}_N)} Z[\cdots, a_i, \cdots] \omega^{a_i \cup A_i}, \tag{D.5}$$

where $I$ is the identity matrix, $J_{i,i} = 1$ and 0 otherwise. $\mathrm{S}_i$ is the EM duality between the $i$-th $\mathbb{Z}_N$ in the bulk and corresponds to gauging the $i$-th $\mathbb{Z}_N$ symmetry on the boundary, i.e. $\mathrm{S}_i$ corresponds to the Kramers-Wannier duality of the $i$-th $\mathbb{Z}_N$ on the boundary.

Note that both R-type and T-type correspond to action with finite depth local unitary on the boundary theory. To classify the non-local mapping on the boundary theory, one should mod out the R-type and T-type transformations [154].

In particular, the triality transformation Tri corresponds to the bulk symmetry,

$$V_{\mathrm{Tri}} = \mathrm{R}(U_1) \cdot \mathrm{S}_2 \cdot \mathrm{S}_1 \cdot \mathrm{T} = \begin{pmatrix} 0 & 0 & 0 & 1 \\ 0 & 0 & -1 & 0 \\ 0 & 1 & -1 & 0 \\ -1 & 0 & 0 & -1 \end{pmatrix}, \tag{D.6}$$

where $U_1 = \begin{pmatrix} 0 & 1 \\ -1 & 0 \end{pmatrix}$. It is easy to check $V_{\mathrm{Tri}}^3 = \mathbb{1}$.

The $p$-ality transformation P is given by,

$$V_{\mathrm{P}} = \mathrm{R}(U_1) \cdot \mathrm{S}_2 \cdot \mathrm{S}_1 \cdot \mathrm{T}^{-2} = \begin{pmatrix} 0 & 0 & 0 & 1 \\ 0 & 0 & -1 & 0 \\ 0 & 1 & 2 & 0 \\ -1 & 0 & 0 & 2 \end{pmatrix}, \tag{D.7}$$

where $U_1 = \begin{pmatrix} 0 & 1 \\ -1 & 0 \end{pmatrix}$ is the same as above. It is easy to check $V_{\mathrm{P}}^p = \mathbb{1} \mod p$, where $p$ is a prime number.

# E Defect

One can obtain the Kramer-Wannier duality defect Hamiltonian using the half-gauging on the lattice as illustrated in [21,93]. We will show the defect Hamiltonian with various (un)twisted half-gaugings.

## E.1 Untwisted Half-gauging $\mathbb{Z}_N \times \mathbb{Z}_N$

We consider half gauging the system $< 2j+1$, then it creates the defect located at $(2j, 2j+1)$. The defect Hamiltonian is closely related to the Kramer-Wannier duality defect and corresponds to doing Kramer-Wannier duality on even and odd sites $\text{KW}^e\text{KW}^o$ and then shift $j \to j - \frac{1}{2}$,

$$
\begin{aligned}
Z_{2j}Z_{2j+2}^{-1} &\to X_{2j}Z_{2j+2}^{-1}, & X_{2j} &\to Z_{2j-2}Z_{2j}^{-1}, \\
Z_{2j-1}Z_{2j+1}^{-1} &\to X_{2j-1}Z_{2j+1}^{-1}, & X_{2j-1} &\to Z_{2j-3}Z_{2j-1}^{-1}.
\end{aligned}
\tag{E.1}
$$

The defect terms for the SPT$^a$ are given by,

$$
\begin{aligned}
Z_{2j-1}^{-a}X_{2j}Z_{2j+1}^a &\to Z_{2j-2}X_{2j-1}^{-a}Z_{2j}^{-1}Z_{2j+1}^a, \\
Z_{2j}^a X_{2j+1}Z_{2j+2}^{-a} &\to X_{2j}^a X_{2j+1}Z_{2j+2}^{-a}.
\end{aligned}
\tag{E.2}
$$

When $N = 2$, such defect Hamiltonian under the renormalization group flow will correspond to inserting the duality defect of $\text{Rep}(H_8)$. A closely related duality defect of $\text{Rep}(D_8)$ is given by $T\text{KW}^e\text{KW}^o$, whose $\mathbb{Z}_N \times \mathbb{Z}_N$ version is,

$$
\begin{aligned}
Z_{2j}Z_{2j+2}^{-1} &\to \widetilde{X}_{2j+1}Z_{2j+2}^{-1}, & X_{2j} &\to Z_{2j-1}\widetilde{Z}_{2j+1}^{-1}, \\
Z_{2j-1}Z_{2j+1}^{-1} &\to X_{2j}Z_{2j+1}^{-1}, & X_{2j-1} &\to Z_{2j-2}Z_{2j}^{-1}.
\end{aligned}
\tag{E.3}
$$

The defect terms for the SPT$^a$ are given by,

$$
\begin{aligned}
Z_{2j-1}^{-a}X_{2j}Z_{2j+1}^a &\to Z_{2j-1}X_{2j}^{-a}\widetilde{Z}_{2j+1}^{-1}Z_{2j+1}^a, \\
Z_{2j}^a X_{2j+1}Z_{2j+2}^{-a} &\to \widetilde{X}_{2j+1}^a X_{2j+1}Z_{2j+2}^{-a}.
\end{aligned}
\tag{E.4}
$$

There is an additional site in the Hilbert space and the dimension of the defect Hilbert space is larger by $N$ times. This corresponds to the quantum dimension of the duality defect.

## E.2 Twisted half-gauging

The twisted half-gauging can be obtained by first applying SPT entangler to half of the space and then applying untwisted half-gauging,

$$
\begin{aligned}
Z_{2j}Z_{2j+2}^{-1} &\to X_{2j}Z_{2j+2}^{-1}, & X_{2j} &\to Z_{2j-2}X_{2j-1}^{-b}Z_{2j}^{-1}, \\
Z_{2j-1}Z_{2j+1}^{-1} &\to X_{2j-1}Z_{2j+1}^{-1}, & X_{2j-1} &\to Z_{2j-3}X_{2j-2}^{b}Z_{2j-1}^{-1}.
\end{aligned}
\tag{E.5}
$$

The defect terms for the SPT$^a$ are given by,

$$
\begin{aligned}
Z_{2j-1}^{-a}X_{2j}Z_{2j+1}^a &\to Z_{2j-2}X_{2j-1}^{-a-b}Z_{2j}^{-1}Z_{2j+1}^a, \\
Z_{2j}^a X_{2j+1}Z_{2j+2}^{-a} &\to X_{2j}^a X_{2j+1}Z_{2j+2}^{-a}.
\end{aligned}
\tag{E.6}
$$

Combined with the translation operator, the defect terms become,

$$
\begin{aligned}
Z_{2j}Z_{2j+2}^{-1} &\to \widetilde{X}_{2j+1}Z_{2j+2}^{-1}, & X_{2j} &\to Z_{2j-1}X_{2j}^{-b}\widetilde{Z}_{2j+1}^{-1}, \\
Z_{2j-1}Z_{2j+1}^{-1} &\to X_{2j}Z_{2j+1}^{-1}, & X_{2j-1} &\to Z_{2j-2}X_{2j-1}^{b}Z_{2j}^{-1}.
\end{aligned}
\tag{E.7}
$$

The defect terms for the SPT$^a$ are given by,

$$Z_{2j-1}^{-a} X_{2j} Z_{2j+1}^{a} \rightarrow Z_{2j-1} X_{2j}^{-a-b} \widetilde{Z}_{2j+1}^{-1} Z_{2j+1}^{a},$$
$$Z_{2j}^{a} X_{2j+1} Z_{2j+2}^{-a} \rightarrow \widetilde{X}_{2j+1}^{a} X_{2j+1} Z_{2j+2}^{-a}. \tag{E.8}$$

Again, the dimension of the defect Hilbert space is $N$ times larger than the original Hilbert space. This corresponds to the quantum dimension of the triality defect.

### E.3 Derivation of half gauging

#### E.3.1 Untwisted half-gauging

KW$^e$KW$^o$

$$Z_{2j} \widetilde{Z}_{2j+\frac{1}{2}} Z_{2j+2}^{-1} \rightarrow \widetilde{X}_{2j+\frac{1}{2}} Z_{2j+2}^{-1}, \qquad X_{2j} \rightarrow \widetilde{Z}_{2j-\frac{3}{2}} \widetilde{Z}_{2j+\frac{1}{2}}^{-1},$$
$$Z_{2j-1} \widetilde{Z}_{2j-\frac{1}{2}} Z_{2j+1}^{-1} \rightarrow \widetilde{X}_{2j-\frac{1}{2}} Z_{2j+1}^{-1}, \qquad X_{2j-1} \rightarrow \widetilde{Z}_{2j-\frac{5}{2}} \widetilde{Z}_{2j-\frac{1}{2}}^{-1}. \tag{E.9}$$

The defect terms for the SPT$^a$ are given by,

$$Z_{2j-1}^{-a} \widetilde{Z}_{2j-\frac{1}{2}}^{-a} X_{2j} Z_{2j+1}^{a} \rightarrow \widetilde{Z}_{2j-\frac{3}{2}} \widetilde{X}_{2j-\frac{1}{2}}^{-a} \widetilde{Z}_{2j+\frac{1}{2}}^{-1} Z_{2j+1}^{a},$$
$$Z_{2j}^{a} \widetilde{Z}_{2j+\frac{1}{2}}^{a} X_{2j+1} Z_{2j+2}^{-a} \rightarrow \widetilde{X}_{2j+\frac{1}{2}}^{a} X_{2j+1} Z_{2j+2}^{-a}. \tag{E.10}$$

#### E.3.2 SPT entangler

$U = \prod_j \mathsf{CZ}_{2j-1,2j}^{-b} \mathsf{CZ}_{2j,2j+1}^{b}$ acts $< 2j + 1$, only the following terms will be modified

$$X_{2j} \rightarrow Z_{2j-1}^{-b} X_{2j}, \tag{E.11}$$
$$Z_{2j-1}^{-a} X_{2j} Z_{2j+1}^{a} \rightarrow Z_{2j-1}^{-a-b} X_{2j} Z_{2j+1}^{a}. \tag{E.12}$$

#### E.3.3 Twisted half-gauging

Combining the SPT entangler and untwisted half-gauging, we have,

$$Z_{2j} \widetilde{Z}_{2j+\frac{1}{2}} Z_{2j+2}^{-1} \rightarrow \widetilde{X}_{2j+\frac{1}{2}} Z_{2j+2}^{-1}, \qquad X_{2j} \rightarrow Z_{2j-1}^{-b} X_{2j} \rightarrow Z_{2j-1}^{-b} \widetilde{Z}_{2j-\frac{1}{2}}^{-b} X_{2j} \rightarrow \widetilde{Z}_{2j-\frac{3}{2}} \widetilde{X}_{2j-\frac{1}{2}}^{-b} \widetilde{Z}_{2j+\frac{1}{2}}^{-1},$$
$$Z_{2j-1} \widetilde{Z}_{2j-\frac{1}{2}} Z_{2j+1}^{-1} \rightarrow \widetilde{X}_{2j-\frac{1}{2}} Z_{2j+1}^{-1}, \qquad X_{2j-1} \rightarrow Z_{2j-2}^{b} X_{2j-1} Z_{2j}^{-b} \rightarrow \widetilde{Z}_{2j-\frac{5}{2}} \widetilde{X}_{2j-\frac{3}{2}}^{b} \widetilde{Z}_{2j-\frac{1}{2}}^{-1}. \tag{E.13}$$

The defect terms for the SPT$^a$ are given by,

$$Z_{2j-1}^{-a} X_{2j} Z_{2j+1}^{a} \rightarrow Z_{2j-1}^{-a-b} X_{2j} Z_{2j+1}^{a} \rightarrow Z_{2j-1}^{-a-b} \widetilde{Z}_{2j-\frac{1}{2}}^{-a-b} X_{2j} Z_{2j+1}^{a} \rightarrow \widetilde{Z}_{2j-\frac{3}{2}} \widetilde{X}_{2j-\frac{1}{2}}^{-a-b} \widetilde{Z}_{2j+\frac{1}{2}}^{-1} Z_{2j+1}^{a},$$
$$Z_{2j}^{a} \widetilde{Z}_{2j+\frac{1}{2}}^{a} X_{2j+1} Z_{2j+2}^{-a} \rightarrow \widetilde{X}_{2j+\frac{1}{2}}^{a} X_{2j+1} Z_{2j+2}^{-a}. \tag{E.14}$$

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
