# Peer review of "Realizing triality and $p$-ality by lattice twisted gauging in (1+1)d quantum spin systems"

_SciPost Physics, doi:SciPost Phys. 17, 136 (2024)_

## Round 1 · Referee Report · Anonymous (Referee 1) · 2024-10-2

Report

The revised manuscript addresses most of the concerns raised by the referees. I am still confused by something about the maps in (3.13) and the related circuit in (3.15).

First, the ordering of operators is not consistent with the flow of time in the circuits. For example, around (3.12), it is mentioned that $U_\mathrm{initial}$ is applied first, which means time flows from bottom to top in (3.15). However, the authors write the associated operator as $U_\mathrm{initial} U_\mathrm{gauge}$, which means $U_\mathrm{gauge}$ is applied first. This inconsistency exists in other equations too and the authors should rectify this.

More importantly, the right-most map in (3.13) does not make sense to me. The action of the circuit $U_\mathrm{gauge}$ on the matter field $Z_i$ is given by the third map. This map already implies the action of the circuit on $Z_i Z_j$ and it gives the expected minimal coupling answer. But, in their response, the authors say that the last two maps together imply this action. Can they please explain why they need the fourth map for this?

It seems like the action of $U_\mathrm{gauge}$ on $\tilde Z_{i+\frac12}$ is given by $\tilde Z_{i+\frac12} \rightarrow \tilde Z_{i+\frac12} \tilde Z_{i+\frac32} \cdots$. What is the interpretation of this in terms of gauging?

Recommendation

Publish (easily meets expectations and criteria for this Journal; among top 50%)

  • validity: top
  • significance: high
  • originality: high
  • clarity: high
  • formatting: perfect
  • grammar: excellent

Author:  Da-Chuan Lu  on 2024-10-04  [id 4831]

(in reply to Report 1 on 2024-10-02)
Category:
answer to question
correction

We thank the referee for clarifying the questions. We agree that the fourth map is unnecessary and have updated our manuscript accordingly. For the sequence of the action of U, we read it from left to right, since we previously defined the transformation of operators as $O\rightarrow O'= U^\dagger O U$.

For the action of $U_{gauge}$ on $\tilde{Z}_{i+1/2}$, $\tilde{Z}_{i+1/2}$ cannot appear alone, it must attached to a $Z_i$ operator. Because in the enlarged Hilbert space, we only introduce $\tilde{X}_{i+1/2}=1,\forall i$ but no $\tilde{Z}_{i+1/2}$. When there is a $Z_i$ operator, the gauge string will continue to infinity. As discussed around (3.26), the additional site 0 with operator $Z_0$ that labels the boundary condition of the Ising chain, under the KW transformation, $Z_0$ becomes the product of $X$ operator which corresponds to the symmetry operator in the dual model (The product of $X$ is the product of $\tilde{Z}$ after some basis transformation that is used to match with the original model).

Attachment:

triality_defect_lattice_10-4.pdf

---

## Round 1 · Referee Report · Anonymous (Referee 3) · 2024-10-3

Report

The authors have addressed my questions in the resubmission. I therefore recommend the publication of this manuscript.

Recommendation

Publish (meets expectations and criteria for this Journal)

---

## Round 1 · Referee Report · Anonymous (Referee 2) · 2024-10-5

Report

The authors address the issue I pointed out in my last report. Therefore, I recommend the publication of the paper.

Recommendation

Publish (meets expectations and criteria for this Journal)

---

## Round 1 · Author Response

We thank the referees for carefully reading our manuscript and expressing a positive evaluation of its contribution to the field. We appreciate that the referees generally agree that our work deserves publication in SciPost.

We answered the referees' questions and updated our manuscript accordingly. We hope the modified version of the manuscript is satisfactory.

---

## Round 1 · List of Changes

We addressed all the referees' questions and fixed several typos in our updated manuscript. In particular, 1. We add footnote 1 on page 2 to discuss the case when gauging the non-abelian symmetry. 2. We add a discussion of $(S_3\times S_3)\rtimes Z_2$ with explicit correspondence between group elements and the mappings on the lattice Hamiltonian.

---

## Editorial Decision

published